# Structural basis for transcription initiation by bacterial ECF σ factors

Lingting Li[1,2], Chengli Fang[1,2], Ningning Zhuang[1], Tiantian Wang[1,2] & Yu Zhang [1]

Bacterial RNA polymerase employs extra-cytoplasmic function (ECF) σ factors to regulate context-specific gene expression programs. Despite being the most abundant and divergent σ factor class, the structural basis of ECF σ factor-mediated transcription initiation remains unknown. Here, we determine a crystal structure of *Mycobacterium tuberculosis* (*Mtb*) RNAP holoenzyme comprising an RNAP core enzyme and the ECF σ factor σH (σH-RNAP) at 2.7 Å, and solve another crystal structure of a transcription initiation complex of *Mtb* σH-RNAP (σH-RPo) comprising promoter DNA and an RNA primer at 2.8 Å. The two structures together reveal the interactions between σH and RNAP that are essential for σH-RNAP holoenzyme assembly as well as the interactions between σH-RNAP and promoter DNA responsible for stringent promoter recognition and for promoter unwinding. Our study establishes that ECF σ factors and primary σ factors employ distinct mechanisms for promoter recognition and for promoter unwinding.

[1] Key Laboratory of Synthetic Biology, CAS Center for Excellence in Molecular Plant Sciences, Shanghai Institute of Plant Physiology and Ecology, Chinese Academy of Sciences, Shanghai 200032, China. [2] University of Chinese Academy of Sciences, Beijing 100049, China. These authors contributed equally: Lingting Li, Chengli Fang.  Correspondence and requests for materials should be addressed to Y.Z. (email: yzhang@sippe.ac.cn)

Transcription initiation is the first and the most tightly regulated step of bacterial gene expression[1–3]. σ factors are required for transcription initiation[4]. After forming a complex with the RNA polymerase (RNAP) core enzyme, σ factors guide RNAP to promoter DNA, open double-stranded DNA (dsDNA) to form a transcription bubble, facilitate synthesis of initial short RNA transcripts, and later assist in promoter escape[4–6].

Bacterial σ factors are classified into two types—σ70- and σ54-type factors based on their distinct structures and mechanisms. The σ70-type factors can be further classified into four groups according to their conserved domains[4]. Group-1 σ factors (or primary σ factors) contain domains $\sigma_{1.1}$, $\sigma_{1.2}$, $\sigma_{NCR}$, $\sigma_2$, $\sigma_{3.1}$, $\sigma_{3.2}$, and $\sigma_4$; group-2 σ factors contain all domains except $\sigma_{1.1}$; group-3 σ factors contain $\sigma_2$, $\sigma_{3.1}$, $\sigma_{3.2}$, and $\sigma_4$; while group-4 or extra-cytoplasmic function (ECF) σ factors only contain $\sigma_2$ and $\sigma_4$[7]. The genomes of a majority of bacteria harbor one primary σ factor for expression of most genes (i.e., group-1 σ factor; σ70 in *Escherichia coli* and σA in Gram-positive bacteria; referred as σA hereafter), and multiple alternative σ factors for expression of genes with cellular- or environmental-context-dependent functions[8,9]. The ECF σ factors are the largest family of alternative σ factors. On average, bacterial genomes encode six ECF σ factors; the given number for a particular bacterium will vary according to its genome size and environmental complexity[9,10]. ECF σ factors enable bacteria to rapidly respond to a variety of stresses[9,11,12] and are known to be essential for the pathogenicity of several disease-causing bacteria[13,14]. *Mycobacterium tuberculosis* has 10 ECF σ factors (σC, σD, σE, σG, σH, σI, σJ, σK, σL, and σM); deletion of ECF σ factors from *M. tuberculosis* results in attenuated disease progression (e.g., *sigC* and *sigD*) or in alleviated virulence (e.g., *sigE* and *sigH*)[15,16].

The σA is capable of recognizing at least five conserved functional elements in the DNA sequences of gene promoters, including the "−35 element" (TTGACA)[17], the "Z element"[18], the "extended −10 element" (TG)[19], the "−10 element" (TATAAT)[17], and the "discriminator element" (GGG)[20]. Distinct domains of σA are responsible for interacting with these DNA elements: the domain $\sigma_4$ forms sequence-specific interactions with exposed bases in the major groove of the −35 dsDNA[21]; the $\sigma_{2.5}$ and $\sigma_{3.1}$ domains reach into the major groove of the extended −10 element and make base-specific contacts[22,23]; and the $\sigma_2$ and $\sigma_{1.2}$ domains recognize and then unwind the −10 element dsDNA[3,24].

During the process of promoter unwinding, a tryptophan dyad of $\sigma_2$ (W256/W257 in *Thermus aquaticus* σA or W433/W434 in *E. coli* σ70) forms a chair-like structure that functions as a wedge to separate the dsDNA at the (−12)/(−11) junction[22,23]. The group-2 σ factors use the same set of residues to unwind promoter DNA; but the melting residues of group-3 σ factors are not conserved[25]. Subsequently, the base moieties of the unwound nucleotides at position −11 and −7 of the nontemplate strand—$A_{(−11)}$(nt) and $T_{(−7)}$(nt)—are flipped out and inserted into pre-formed pockets by $\sigma_2$ and $\sigma_{1.2}$[3,24]. Domain $\sigma_{1.2}$ also recognizes the discriminator element by flipping out the guanine base of $G_{(−6)}$(nt) and inserting it into a pocket[24]. Although $\sigma_{3.2}$ does not read the promoter sequence directly, it is essential for transcription initiation. Domain $\sigma_{3.2}$ reaches into the RNAP active site cleft and "pre-organizes" template single-stranded DNA (ssDNA)[24]. Domain $\sigma_{3.2}$ also blocks the path of the extending RNA chain (>5 nt)[26,27] thereby contributing to both initial transcription pausing[28] and promoter escape[29,30].

Each category of known ECF σ factors recognizes promoters bearing a unique sequence signature at the −35 and the −10 elements[10,31]. In contrast to the high tolerance to sequence variation at the −35 and the −10 promoter elements exhibited by the primary σ factor, the ECF σ factors have stringent requirements for sequence identity in the −35 and the −10 elements and for spacer length between these two elements through an unknown mechanism[8,32]. Although both the primary and ECF σ factors recognize the −35 element via $\sigma_4$ and recognize the −10 element via $\sigma_2$, the protein sequences of these two domains are not well conserved, and the consensus sequences of the two corresponding DNA elements vary. Crystal structures of individual $\sigma_2$ or $\sigma_4$ domains of ECF σ factors complexed with cognate DNA have suggested that these ECF domains bind the −35 and the −10 elements differently than does the primary σ factor, implicating a unique means of promoter recognition by ECF σ factors[33,34]. Another striking difference was revealed by a sequence analysis showing the surprising fact that ECF σ factors do not contain $\sigma_3$ domains ($\sigma_{3.1}$ and $\sigma_{3.2}$), but instead contains a linker—highly variable in both length and sequence—to connect the $\sigma_2$ and $\sigma_4$ domains[4]. This fact immediately raises the question of how these σ factors perform multiple steps of transcription initiation that the $\sigma_3$ domain performs in the primary σ factor.

A recent crystal structure of *E. coli* $\sigma^E_2$/−10 ssDNA binary complex suggests that bacterial ECF σ factors probably recognize and unwind promoters through a unique mechanism. Specifically, *E. coli* σE employs a flexible "specificity loop" to recognize a flipped master nucleotide of the −10 element and probably unwinds at a distinct position compared with that of σ70 by using non-conserved melting residues[34]. In contrast to the large collection of structural information of primary σ factor, no structure of bacterial RNAP complex with ECF σ factor is available. Therefore, it is largely unknown how ECF σ factors form a holoenzyme with RNAP and how ECF σ factors work alongside RNAP to recognize and to unwind promoter DNA. Here we report the crystal structure of an ECF σ factor-RNAP holoenzyme comprising *M. tuberculosis* RNAP and σH at 2.70 Å resolution. We also report the crystal structure of an ECF σ factor-RNAP transcription initiation complex comprising *M. tuberculosis* σH-RNAP holoenzyme, a full transcription bubble of promoter DNA, and an RNA primer at 2.80 Å resolution. The crystal structures present detailed interactions among RNAP, ECF σ factors, and promoter DNA. The structures together with data from biochemical assays collectively establish the structural basis of RNAP holoenzyme assembly, promoter recognition, and promoter unwinding by the ECF σ factors.

## Results

**The crystal structure of *M. tuberculosis* σH-RNAP holoenzyme.** The crystals of *M. tuberculosis* σH-RNAP holoenzyme were unexpectedly obtained during an initial attempt to crystallize *M. tuberculosis* σH-RPo (Supplementary Figs. 1a, c–f). The crystal structure of *M. tuberculosis* σH-RNAP holoenzyme at 2.7 Å resolution was determined by molecular replacement using a *Mycobacterium smegmatis* RNAP core enzyme (PDB: 5TW1) [https://www.rcsb.org/structure/5TW1] as the searching model[35]. The Fo–Fc map shows unambiguous density for σH residues 22–195 (Table 1; Supplementary Fig. 2a) and the anomalous difference map shows clear density for 4 out of 5 Se atoms, validating the σH model (Supplementary Fig. 2a).

$\sigma^H_2$ (residues 22–99) and $\sigma^H_4$ (residues 140–195) fold into independent helical domains (Fig. 1a, c). Lacking the $\sigma_{1.1}$, $\sigma_{1.2}$, and $\sigma_{NCR}$ domains of σA, the $\sigma^H_2$ domain is very compact, containing only four α helices (Fig. 1c). The "specificity loop" (residues 72–79 in $\sigma^H_2$; Supplementary Fig. 2a) known to be essential for recognition of the −10 element is disordered (no electron density), in contrast to the pre-organized specificity loop in the σA-RNAP holoenzyme (Fig. 1b, d and Supplementary Fig. 2b–c). Lacking $\sigma_{1.2}$, the domain that forms extensive

**Table 1 The statistics of crystal structures**

|  | *M. tuberculosis* σH-RNAP | *M. tuberculosis* σH-RPo |
|---|---|---|
| *Data collection* |  |  |
| Space group | $P2_1$ | $P2_12_12_1$ |
| Cell dimensions |  |  |
| $a, b, c$ (Å) | 130.6, 159.8, 131.4 | 129.8, 164.0, 214.8 |
| $\alpha, \beta, \gamma$ (°) | 90, 119, 90 | 90, 90, 90 |
| Resolution (Å) | 50.00–2.75 | 50.00–2.80 |
|  | (2.80–2.75) | (2.85–2.80) |
| $R_{sym}$ or $R_{merge}$ | 0.078 (0.893) | 0.164 (1.430) |
| $I/\sigma I$ | 21.4 (1.2) | 12.9 (1.2) |
| Completeness (%) | 98.2 (84.3) | 99.8 (99.5) |
| Redundancy | 5.4 (3.5) | 8.5 (8.4) |
| $CC_{1/2}$ in highest shell | 0.530 | 0.547 |
| *Refinement* |  |  |
| Resolution (Å) | 50.00–2.75 | 50.00–2.80 |
| No. of reflections | 119,643 | 112,969 |
| $R_{work}/R_{free}$ | 0.218/0.258 | 0.220/0.255 |
| No. of atoms | 23,311 | 25,858 |
| $B$-factors (Å$^2$) | 90.5 | 66.1 |
| R.m.s deviations |  |  |
| Bond lengths (Å) | 0.003 | 0.005 |
| Bond angles (°) | 0.565 | 0.621 |
| Ramachandran plot |  |  |
| Favored (%) | 97.63 | 97.40 |
| Allowed (%) | 2.37 | 2.60 |
| Disallowed (%) | 0 | 0 |

Numbers in parenthesis are for the highest resolution

interactions with the specificity loop of σA, probably accounts for the disordered conformation of the specificity loop in σH (Fig. 1c, d). As occurs in σA2, the σH2 domain resides in a cleft between the RNAP-β lobe and the RNAP-β′ coiled-coil (β′CC) and makes extensive electrostatic interactions with the latter (Fig. 1e). Notably, the residues contacting β′CC of both σA and ECF σ factors are conserved (Supplementary Figs. 2d, e and 3), suggesting that β′CC probably serves as an anchor point for the σ2 domain of the majority of bacterial σ factors.

The σH4 domain enfolds the flap-tip helix of the RNAP-β subunit (βFTH; Figs. 1e and 2a). The hydrophobic residues contacting the βFTH are conserved between the σA and the ECF σ factors (Supplementary Figs. 2f–g and 3). Surprisingly, we discovered another anchor point for the σH4 domain on RNAP —a C-terminal helix of the RNAP-β subunit (βCTH; residues 1145–1157; Figs. 1e and 2a, and Supplementary Fig. 2h and m). The interaction with βCTH was not observed in any of the previously reported bacterial σA-RNAP structures[22,24,36,37]. To explore the contribution of such interaction to the transcription activity of σH-RNAP, we performed in vitro transcription experiments using wild-type or βCTH-deleted *Mtb* σH-RNAP holoenzyme and *pClpB* promoter variants with −35/−10 spacer lengths ranging from 15 to 19 bp. The wild-type σH-RNAP was most transcriptionally active with a promoter of 17-bp spacer (Fig. 2b), consistent with a study reporting that most σH-regulated promoters have a 17-bp spacer[38]. The βCTH-deletion variant caused impaired transcription activity from promoter with the optimal spacer length (17 bp) but showed little effect on promoter with sub-optimal spacer lengths (16 and 18 bp) (Fig. 2b), suggesting that the interactions between βCTH and σH are important for the transcription activity of σH. Intriguingly, deletion of βCTH caused a general increase of σA-dependent transcription activity from promoter with spacer lengths 15–19 bp (Supplementary Fig. 2i).

The most surprising finding in the structure of σH-RNAP is the interaction between RNAP and the linker region connecting σH2

and σH4. The linker is the least conserved region among the ECF σ factors and shares no sequence similarity with the linker of σA (Supplementary Fig. 3). Our structure shows that the linker region of σH dives into the active site cleft and emerges out from the RNA exit channel of RNAP (Figs. 1e and 2c, and Supplementary Fig. 4a–b). This interaction creates an entry channel for template ssDNA loading into the active site cleft during RPo formation, but blocks the exit pathway of extending RNA during subsequent transcription initiation events. The path of the σH2/σH4 linker in RNAP is similar to that of σA3.2 (Supplementary Fig. 4c–d), so we designated the σH2/σH4 linker as σH3.2.

To examine the significance of the interaction between σH3.2 and RNAP, we tested the in vitro transcription activity of RNAP holoenzyme comprising σH variants with the σH3.2 domain either deleted or swapped. Deleting σH3.2 or replacing σH3.2 with a protein sequence known to be disordered completely abolished the transcription activity of σH (lanes II and III in Fig. 2d), indicating that σH3.2 does not simply serve as a σH2/σH4 linker; rather, the interaction between σH3.2 and the active site cleft of RNAP is essential for transcription. Interestingly, replacing σH3.2 with the σ2/σ4 linker from other ECF σ factor partially recovered the transcription activity (lanes IV, V, and VI in Fig. 2d and Supplementary Fig. 4e). Based on these results, we infer that other ECF σ factors probably have a functional σ3.2 domain that, while divergent in sequence, likely binds RNAP in a way somehow analogous to σH3.2.

**The overall structure of *Mtb* σH-RPo.** To understand how σH-RNAP holoenzyme recognizes promoter DNA and initiates transcription, we sought to determine a crystal structure of σH-RPo. We assembled the complex by incubating the RNAP core enzyme, σH, and a synthetic nucleic-acid scaffold (Fig. 3d, and Supplementary Fig. 1b, 1g–j). The synthetic scaffold comprises an upstream DNA duplex (−34 to −10 with respect to transcription start site at +1) with a consensus −35 element (GGAACA), a non-complimentary transcription bubble (−9 to +2) with a consensus −10 element (GTT), a 7-nt RNA primer complimentary to template DNA (−6 to +1), and a downstream DNA duplex (+3 to +13). We determined the crystal structure of σH-RPo at 2.8 Å resolution by molecular replacement using the crystal structure of *Mtb* σH-RNAP holoenzyme as a search model (Table 1). The Fo–Fc map contoured at 2.5 σ shows clear density for all nucleotides of nontemplate ssDNA, template ssDNA, and RNA primer of the transcription bubble, as well as for all the nucleotides of the upstream and downstream DNA duplexes (Fig. 3e).

In the σH-RPo structure, σH makes the same interactions with RNAP as in the structure of σH-RNAP holoenzyme. The RNAP clamp adopts a closed conformation as in σA-RPo[24,39], consistent with previous single-molecule fluorescence resonance energy transfer results[40] and supporting the idea that clamp closure is also an obligatory step of RPo formation in ECF σ-mediated transcription initiation (Fig. 3b). The DNA/RNA hybrid resides in the active site cleft in a post-translocation state, and the downstream DNA duplex is accommodated in the main channel. The conformations of the DNA/RNA hybrid and downstream DNA duplex in σH-RPo and σA-RPo are similar (Fig. 3c)[24].

The σH-RPo structure revealed multiple interactions responsible for promoter recognition and promoter unwinding by σH-RNAP that we will describe in detail in each of the subsequent sections of our manuscript. These include: (1) σH4 inserts into the major groove and reads the sequence of the −35 element (Figs. 3a and 4a, and Supplementary Fig. 6a); (2) the RNAP-β′ subunit stabilizes the upstream DNA duplex by contacting the phosphate backbone of nucleotides at positions −23, −20, and −19 (Figs. 3a

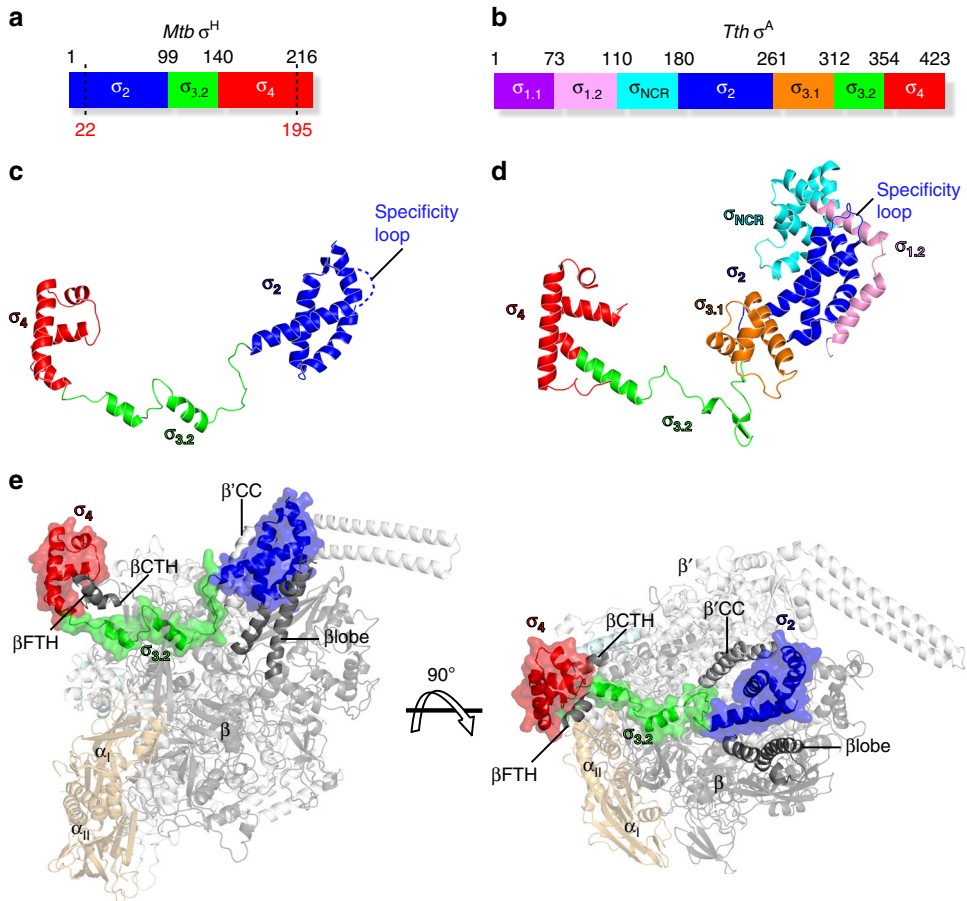

**Fig. 1** The crystal structure of *Mtb* σ^H-RNAP holoenzyme. **a** Schematic diagram of *Mtb* σ^H. Ordered regions in the structure are indicated by dashes. **b** Schematic diagram of *T. thermophilus* σ^A. **c** σ^H in the crystal structure of *Mtb* σ^H-RNAP holoenzyme. The disordered specificity loop is shown by blue dashes. **d** σ^A in the crystal structure of *T. thermophilus* σ^A-RNAP (PDB: 1IW7) [https://www.rcsb.org/structure/1iw7] was used for comparison due to no available structure of *Mtb* σ^A-RNAP holoenzyme. **e** Front and top views of *Mtb* σ^H-RNAP holoenzyme. βFTH, the flap-tip helix on RNAP-β subunit; βCTH, the C-terminal helix on RNAP-β subunit; β'CC, the coiled-coil on RNAP-β' subunit. σ^A_{1.1}, purple; σ^A_{1.2}, pink; σ^A_{NCR}, cyan; σ2, blue; σ^A_{3.1}, orange; σ_{3.2}, green; σ_4, red. RNAP-α subunits, light orange; RNAP-β subunit, black; RNAP-β' subunit, gray; RNAP-ω subunit, light cyan

and 4b); (3) σ^H_2 unwinds dsDNA using an apparently distinct mechanism (Figs. 3a and 4e); (4) σ^H_2 and the RNAP-β subunit recognize sequences at four positions in the −10 element via interactions with nontemplate ssDNA (Figs. 3a and 5a–c); (5) the RNAP-β subunit recognizes the "CRE element" DNA sequence via interactions with nontemplate ssDNA (Figs. 3a and 5d, and Supplementary Fig. 7a); and (6) σ^H_{3.2} guides the template ssDNA into the RNAP active center and forms interactions with the DNA/RNA hybrid in the active site cleft (Figs. 3a and 5e, and Supplementary Fig. 7g).

**The interactions of σ^H_4 with the −35 element.** σ^H-regulated promoters have a distinct consensus sequence at their −35 elements (5′-GGAAYA-3′; from −34 to −29; Supplementary Fig. 5a)[38,40,41]. Alternation of DNA sequences at each of the positions from −34 to −29 resulted in substantial loss of transcription activity (Supplementary Fig. 5b). In the structure, σ^H_4 binds to the major groove of dsDNA of the −35 element and makes base-specific polar interactions with nucleotides at three (−34, −33, and −31) out of six positions (Figs. 3d and 4a, and Supplementary Fig. 6a). The G_{−34}(nt) makes two H-bonds with R186 through its O6 and N7 atoms; the G_{−33}(nt) makes one H-bond with S182 through its O6 atom; and the A_{−31}(nt) makes one H-bond with M181. Moreover, M181 forms extensive van der Waals interactions with nucleotides at positions −31 and −30 of

the template strand. The interactions are important, as alanine substitutions of R186 or S182 resulted in substantial loss of transcription activity (Fig. 4c). Interestingly, the M181A mutant had increased transcription activity, but this came at the apparent expense of relaxing its sequence stringency for the positions −31 and −30 (Fig. 4d), suggesting that M181 partially accounts for sequence specificity of the two positions.

The rest of the −35 element (−32, −30, and −29) makes no base-specific interactions. Previous crystal structure of *E. coli* σ^E_4/−35 dsDNA and a structural model of *Streptomyces coelicolor* σ^R_4/−35 dsDNA reported a local DNA shape readout (straight helix with a narrow minor groove) at this region[33,42]. We observed a similar DNA conformation (Supplementary Fig. 6b) in our crystal structure, suggesting a general mode of promoter recognition for the ECF σ factors. Such DNA structure might assist the binding of −35 dsDNA to the σ^H_4 surface perhaps by making favorable interactions through its phosphate backbones with polar residues of σ^H_4 including Y166, K167, T179, R183, H185, and R188 (Fig. 3d and Supplementary Fig. 6a). Consistently with this, losing any of these interactions causes a substantial loss of transcription activity (Fig. 4c).

**The interactions of σ^H-RNAP with the −35/−10 spacer.** In the crystal structure of *Mtb* σ^H-RPo, σ^H-RNAP contacts phosphate

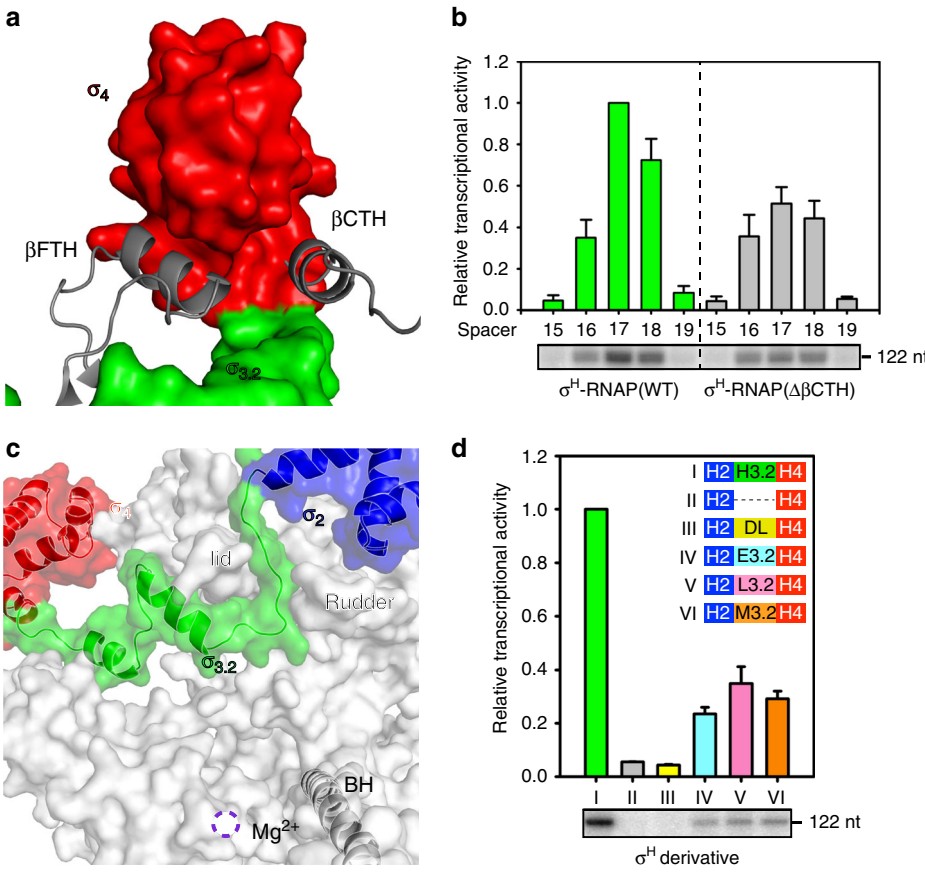

**Fig. 2** The interaction between *Mtb* RNAP core enzyme and σ$^H$. **a** Both βFTH and βCTH interact with σ$^H_4$. Colors are as in previous figure. **b** The in vitro transcription activity of σ$^H$-RNAP(WT) (green bars) and σ$^H$-RNAP(ΔβCTH) (gray bars) from *pClpB* promoter variants with −35/−10 spacer length of 15–19 base pairs. "122 nt" indicates length of run-off RNA products. **c** The interaction between σ$^H_2$/σ$^H_4$ linker (σ$^H_{3.2}$) and RNAP core enzyme. RNAP-β subunit was omitted for clarity. The location of catalytic Mg$^{2+}$ is shown by a dashed purple circle. BH, bridge helix. RNAP-β' subunit, gray. σ$^H_2$, σ$^H_{3.2}$, and σ$^H_4$, blue, green, and red respectively. **d** The in vitro transcription activity from *pClpB* promoter of RNAP holoenzymes comprising σ$^H$ derivatives. H2-H3.2-H4, wild-type σ$^H$; H2---H4, two individual domains of σ$^H_2$ and σ$^H_4$; H2-DL-H4, a chimeric σ$^H$ with σ$^H_{3.2}$ replaced by a disordered loop with an equivalent residue number; H2-E3.2-H4, a chimeric σ$^H$ with σ$^H_{3.2}$ replaced by *Mtb* σ$^E_{3.2}$; H2-L3.2-H4, a chimeric σ$^H$ with σ$^H_{3.2}$ replaced by *Mtb* σ$^L_{3.2}$; H2-M3.2-H4, a chimeric σ$^H$ with σ$^H_{3.2}$ replaced by *Mtb* σ$^M_{3.2}$. The experiments were repeated in triplicate, and the data are presented as mean ± S.E.M. Source data of **b** and **c** are provided as a Source Data file

backbones of the spacer region between the −35 and the −10 elements at three positions (Fig. 3d): (1) R77 of the RNAP-β' zinc-binding domain contacts the nucleotide at position −23; (2) R37 of the RNAP-β' zipper domain contacts the nucleotides at positions −20 and −19; and (3) K96 and R99 of σ$^H_2$ contact the nucleotide at position −14 (Fig. 4b). These interactions probably stabilize the conformation of the upstream DNA duplex, likely promoting the engagement of the upstream duplex with σ$^H_4$ and σ$^H_2$ for subsequent promoter unwinding. Mutating K96 and R99 causes a mild loss of transcription activity, suggesting the importance of these interactions (Fig. 4f).

**The promoter DNA unwinding function of σ$^H$.** The electron density map unambiguously shows that the T:A base pair at position −10 (corresponding to position −12 of promoters for σ$^A$) is unwound, despite the fact that the −10 nucleotides in the synthetic nucleotide scaffold were designed to be complimentary (Figs. 3d and 4e). This observation strongly suggests that σ$^H_2$ unwinds promoter DNA at the −11/−10 junction (corresponding to (−13)/(−12) of promoters targeted by σ$^A$; Fig. 4e and Supplementary Fig. 6c, d). This clearly confirms the hypothesis that the ECF σ factors unwind promoter DNA starting from a distinct position as compared to σ$^A$[32,34]. In the structure, N88 blocks the

pathway of upstream dsDNA and serves as a wedge to disrupt the stacking of base pairs at the positions −11 and −10. The base pair at position −10 is subsequently forced open by σN88 via a competitive H-bond between the Watson-Crick atom of the T$_{−10}$(nt) and N88 (Fig. 4e). Two unwound nucleotides on the nontemplate strand DNA (T$_{−10}$(nt) and T$_{−9}$(nt)) are stabilized by two adjacent pockets of σ$^H_2$; these pockets are where the sequence identities are "read" (Supplementary Fig. 6F). Moreover, two unwound nucleotides on the template strand DNA (A$_{−10}$(t) and T$_{−9}$(t)) are also trapped in a cleft created by the RNAP-β lobe and σ$^H_2$. Specifically, the base moieties of A$_{−10}$(t) and T$_{−9}$(t) form a stack with βR395, σY90, and σY94, and the phosphate moieties are stabilized by βK428, βN419, and σQ98. The functional importance of these residues for promoter DNA unwinding was underscored by our finding that their substitution with alanine resulted in defects in transcription (I85G, I85A, and N88A; Fig. 4f).

Structure superimposition between σ$^H$-RPo and σ$^A$-RPo shows that the promoter DNA position at which unwinding is initiated differs between σ$^A$ and σ$^H$ by one base pair; σ$^H$ unwinds promoter DNA at a position 1 bp upstream of the position at which σ$^A$ unwinds its promoter DNA (the (−12)/(−11) junction for σ$^A$; the −11/−10 junction for σ$^H$ corresponding to the (−13)/(−12) junction for σ$^A$; Supplementary Fig. 6c–e). A tryptophan

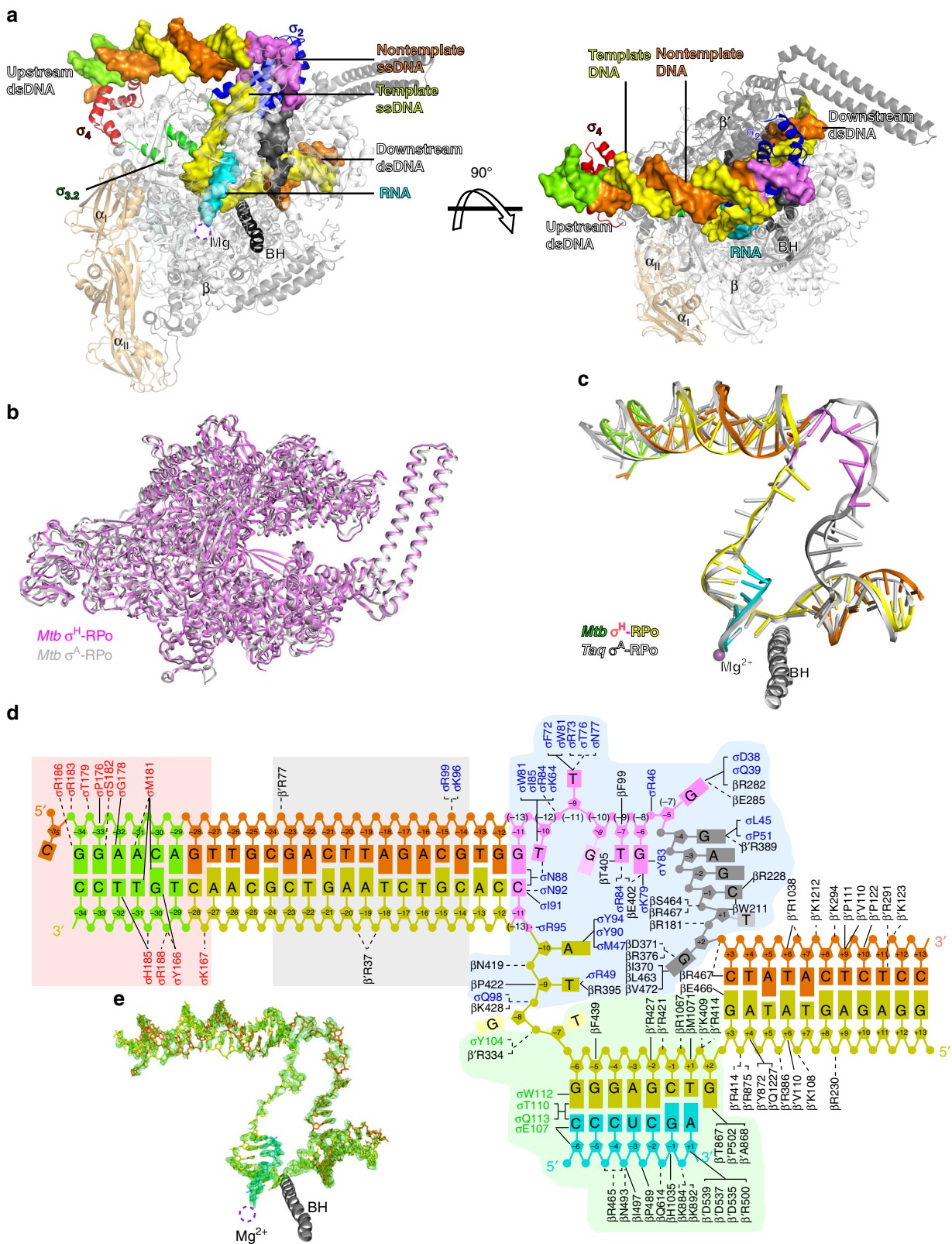

dyad (W433/W434 in *E. coli* or W256/W257 in *T. aquaticus*) is essential for promoter unwinding at the (−12)/(−11) junction by σ[A] [22,23,43,44], but the residues at the corresponding positions of σ[H] (R84/I85) are not conserved (Supplementary Fig. 3). Mutating R84 and I85 in σ[H] to tryptophan (I85W, R84W, or I85W/R84W) resulted in substantial loss of transcription activity, confirming that σ[H] opens promoter through a different mechanism than σ[A] and supporting the mechanism proposed for *E. coli* σ[E] (Fig. 4f)[34].

**The interactions of σ[H]-RNAP with the −10 element.** σ[H]-regulated promoters contain a "G$_{-11}$T$_{-10}$T$_{-9}$" consensus sequence at the −10 element (Supplementary Fig. 5a)[38,40,41].

**Fig. 3** The crystal structure of *Mtb* σ^H-RPo. **a** Front and top views of σ^H-RPo overall structure. The α, β, β′, and ω subunits of RNA polymerase core enzyme are shown as ribbon and colored in light orange, gray, black, and light cyan respectively. The σ^H_2, σ^H_3.2, and σ^H_4 are shown as ribbon and colored as in above figures. The nontemplate DNA, template strand DNA, and RNA strands are shown in surface and colored in orange, yellow, and cyan, respectively, except the −35 element (green), the −10 element (purple), and the CRE (black). The location of the catalytic Mg^2+ is indicated by a dashed circle. **b** Both *Mtb* σ^H-RPo (violet) and *Mtb* σ^A-RPo (light gray; PDB: 5UHA) [https://www.rcsb.org/structure/5uha] show closed clamp conformation. **c** The comparison of upstream double-stranded DNA (dsDNA), transcription bubble, and downstream dsDNA in *Mtb* σ^H-RPo (colored as in **a**) and *Taq* σ^A-RPo (light gray; PDB: 4XLN) [https://www.rcsb.org/structure/4XLN]. **d** Summary of protein–nucleic acid interactions. Solid line, van del Waals interactions; dashed line, polar interactions. Colors are as in above. Red box, interactions with the −35 element (details in Fig. 4a); gray box, interactions with the −35/−10 spacer (details in Fig. 4b); blue box, interactions of the single-stranded DNA (ssDNA) in transcription bubble (details in Figs. 4e and 5a–d); green box, interactions with the DNA/RNA hybrid (details in Fig. 5e). The numbers in parenthesis are corresponding positions in σ^A-regulated promoters. **e** The simulated-annealing omit Fo–Fc electron density map (nucleic acids removed; green; contoured at 2.5 σ) and model for nucleic acids

Alteration of the DNA sequence at any of these positions resulted in complete loss of promoter activity (Supplementary Fig. 5b), helping explain the reported finding from previous bioinformatic studies that the −10 element is the most conserved region among ECF σ factors[10,31]. Our crystal structure shows that the base moiety of $C_{-11}(t)$ of the −11 G:C pair makes one H-bond with N92 and extensive Van der Waal interactions with I91, alanine substitution of N92 or I91 resulted in modest or substantial decrease of transcription, respectively (Fig. 5a, f), providing a structural explanation for sequence recognition at this position. Our crystal structure further reveals that σ^H recognizes the nucleotides of the next two positions via two protein pockets on the surface of σ^H_2 (Fig. 5a).

$T_{-10}(nt)$ is accommodated by a shallow protein pocket on σ^H_2 wherein I85 forms a stack with the base of $T_{-10}(nt)$ at the pocket bottom and W81 supports the sugar moiety of $T_{-10}(nt)$ on one side of the pocket (Fig. 5a). N88 on the other side of the pocket makes a H-bond with the base moiety of $T_{-10}(nt)$, likely contributing to the sequence specificity known to occur for this position. Alanine substitution of I85 or W81 causes severe defects in transcription activity (Figs. 4f and 5f), emphasizing their importance. Sequence alignment of the 10 *Mtb* ECF σ factors shows that the I85 and W81 are highly conserved (Supplementary Fig. 3), suggesting the −10(nt) pocket probably exists on other ECF σ factors. Alanine substitution of N88 causes defects in transcription activity (Fig. 4f), and the sequence alignment shows that N88 is the most frequent residue at this position. However, other polar residues occur at this position (e.g., a histidine for σ^C and σ^D, and an arginine for σ^I, σ^J, and σ^K) (Supplementary Fig. 3), suggesting that this position may help determine sequence specificity for position −10 of the promoter DNA.

$T_{-9}(nt)$ is flipped out and inserted into a protein pocket formed by the "specificity loop" of σ^H_2[34]. The thymine base of $T_{-9}(nt)$ stacked on top of W81 makes one H-bond with the main-chain atom of σ^H residue R73 and two H-bonds with the side-chain atoms of σ residues T76 and N77 on the specificity loop (Fig. 3d and Supplementary Fig. 6f). F72 and R73 contact the thymine base via van der Waals interactions. Mutating $T_{-9}(nt)$ completely abolished promoter activity (Supplementary Fig. 5b), and alanine substitution of σ^H residues contacting $T_{-9}(nt)$ (F72A, T76A, N77A, and W81A) severely decreased transcription activity from the consensus promoter (Fig. 5f), verifying the requirement for the $T_{-9}(nt)$/σ^H_2 interaction for transcription. The fact that both the primary and the ECF σ factors use the specificity loop to read the sequence identity (position −9 of σ^H-regulated promoters corresponding to position (−11) of σ^A-regulated promoters; Supplementary Fig. 6f–h) implies the central importance of this position in promoter DNA[3,24,34]. Outside of these crucial positions, σ^H_2 forms fewer interactions with nontemplate nucleotides (positions −8, −7, and −6). In the structure, the base moiety of nucleotide at position −8 is

disordered (no electron density) and the base moieties of −7 and −6 nucleotides are sandwiched between residue T405 of the RNAP-β subunit and residue Y83 of σ^H_2 (Figs. 3d and 5b, and Supplementary Fig. 6i).

A surprising finding in the σ^H-RPo crystal structure is that σ^H_2 flips the guanine base of $G_{-5}(nt)$ (corresponding to the (−7) position of σ^A-regulated promoters) and inserts into a shallow pocket created by σ^H_2 and the RNAP-β gate loop (Fig. 5c). In this pocket, $G_{-5}(nt)$ forms a stacking interaction with R282, two H-bonds with E285 on the RNAP-β gate loop, and has van der Waals interactions with D38 and Q39 of σ^H_2. Mutations of the RNAP-β gate loop (βR282A or βE285A) cause severely reduced transcription activity (Fig. 5f). As the RNAP-β gate loop makes base-specific contacts to $G_{-5}(nt)$, we tested whether this position exhibits sequence preference. Results of in vitro transcription assays showed that promoters with C or G at this position have much higher transcription activity compared with T or A (Supplementary Fig. 5b), suggesting a sequence preference of C~G > T~A at such position. Our results therefore show that in σ^H-regulated promoters, the consensus sequence of the −10 elements is extended to "$G_{-11}T_{-10}T_{-9}N_{-8}N_{-7}N_{-6}S_{-5}$". It is worth noting that σ^A also accommodates the $T_{(-7)}(nt)$ nucleotide in a protein pocket[3,24]; however, the protein pocket is mainly formed by residues from σ^A_1.2 and σ^A_2[24], and the structural features that determine sequence specificity are located on σ^A (Supplementary Fig. 6i–k).

**Interactions of σ^H-RNAP with the CRE**. The guanine base of $G_{+2}(nt)$ is inserted into the "G" pocket in σ^H-RPo (Fig. 5d) and makes essentially the same interaction with residues in the "G" pocket as the $G_{(+2)}(nt)$ does in the σ^A-RPo complex (Figs. 3d and 5d, and Supplementary Fig. 7d–f)[24]. However, in contrast to the σ^A-RPo, in which the nontemplate $T_{(+1)}(nt)$ forms a stacking interaction with βW211, the nontemplate $T_{+1}(nt)$ in σ^H-RPo was pushed out from the base-stacking. Instead, the nucleotide immediately upstream of +1 makes the stacking interaction with βW211 in σ^H-RPo (Fig. 5d and Supplementary Fig. 7a–c). The base moieties of the remaining nucleotides (−4 to −1) between the CRE and −10 elements are stacked in a row between L45 on σ^H_2 and W211 on the RNAP-β subunit (Fig. 5d and Supplementary Fig. 7a).

**Interactions of the template ssDNA**. In the σ^H-RPo structure, σ^H and the RNAP-β subunit form an "T-ssDNA entry channel" that guides the template ssDNA into the RNAP active center cleft (Supplementary Fig. 4a). Along the channel, σ^H and RNAP form extensive interactions with the template ssDNA (σM47, σR49, σY90, σY94, σQ98, σY04, βR395, βN419, βP422, βK428, and β′R334; Fig. 3d). Compared to σ^A, which forms extensive interactions between the σ^A_3.2 finger with the ssDNA template nucleotides the active center cleft[24], σ^H forms fewer interactions (Fig. 3d

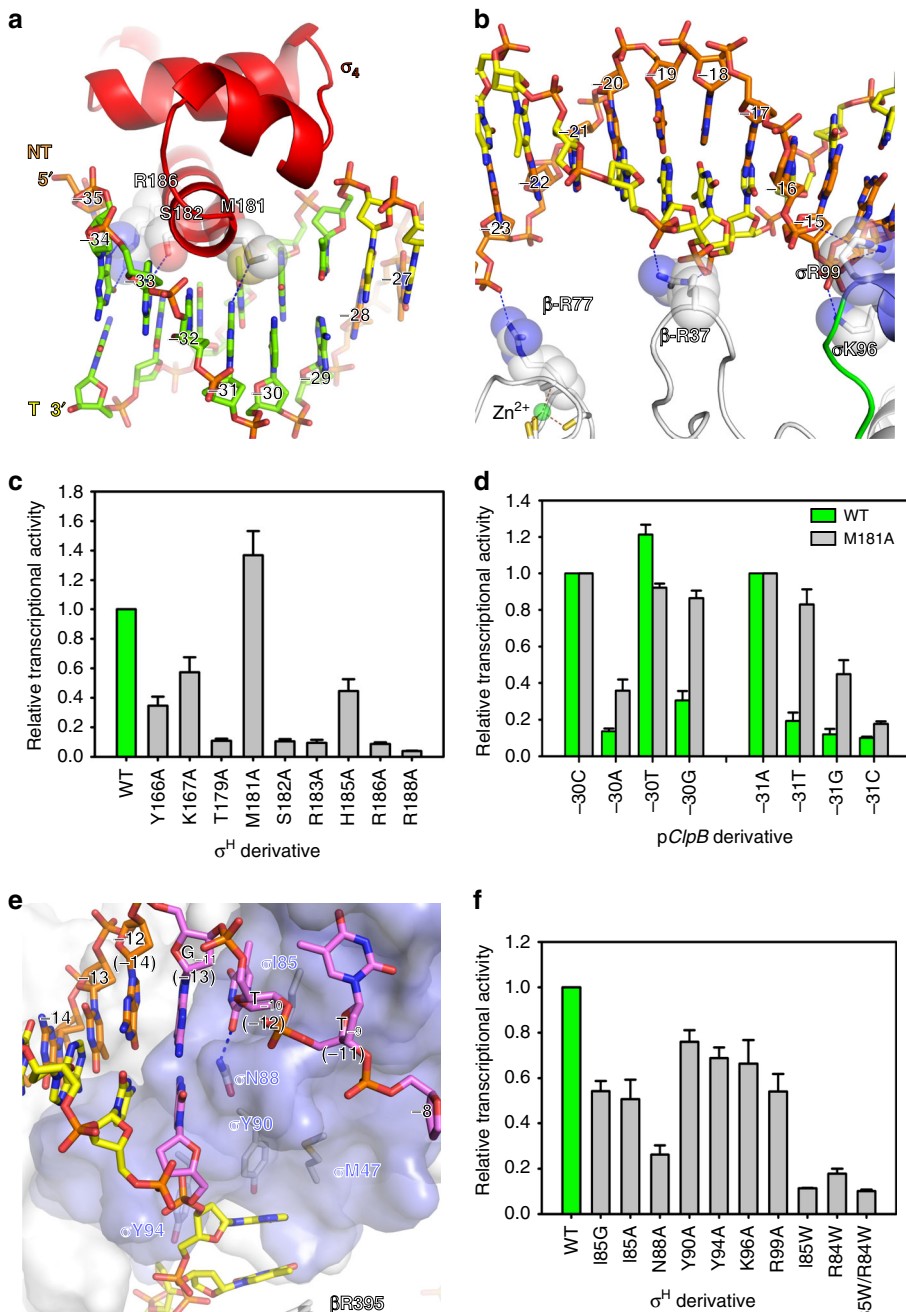

**Fig. 4** The interaction between upstream promoter DNA and RNAP in the *Mtb* σ^H-RPo structure. **a** The interaction between σ^H_4 and the −35 element. σ^H_4, red ribbon. The residues making base-specific interactions are presented as ribbon and half-transparent spheres; the C, O, N, and S atoms of the residues are colored in white, red, blue, and yellow respectively. The C, O, N, and P atoms of the −35 DNA are colored in green, red, blue, and orange, respectively. H-bond, blue dash. **b** The interaction between RNAP and −35/−10 spacer region of upstream dsDNA. The residues making polar interactions with DNA are shown and colored as above. The C atoms of the −35/−10 spacer DNA are colored in yellow (template DNA) or orange (nontemplate DNA), and the rest of atoms are colored as above. **c** The in vitro transcription activity of σ^H derivatives comprising alanine substitution on σ^H_4. **d** M181A changes the sequence specificity for positions −30 and −31 of the −35 element. **e** The promoter melting by σ^H. The numbers in parenthesis correspond to positions in σ^A-RPo. The −10 element is colored in purple. The residues involved in promoter melting are shown in stick and colored in white (C atoms), red (O atoms), and blue (N atoms). **f** The in vitro transcription activity of σ^H_2 derivatives with alanine or tryptophan substitutions. The experiments were repeated in triplicate, and the data are presented as mean ± S.E.M. Source data of **c**, **d**, and **f** are provided as a Source Data file

and Supplementary Fig. 7g–i). The 5′ terminus of the 7-nt RNA is positioned very closely to the tip of σ^H_3.2; extending the RNA by even one additional nucleotide would likely result in steric hindrance (Fig. 5e and Supplementary Fig. 7g). Since σ^H_3.2 occupies the RNA exit channel and must be displaced by the nascent RNA chain, such hindrance may be the trigger for the release of σ^H and promoter DNA during promoter escape.

## Discussion

The structural basis of transcription initiation by the primary σ factor has been studied extensively, but little is known about how the ECF σ factors—the largest and most diverse group of σ^70 family factors—initiate transcription. In this study, we present high-resolution crystal structures of *M. tuberculosis* σ^H-RNAP holoenzyme and σ^H-RPo complexes along with comprehensive

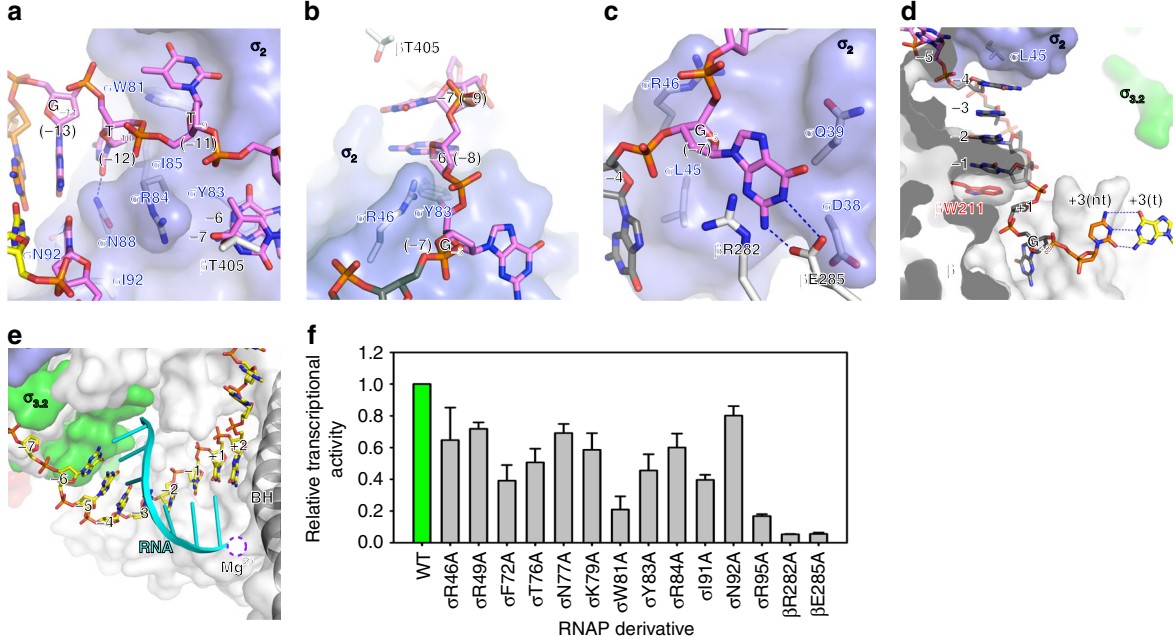

**Fig. 5** The interaction between transcription bubble and RNAP in the *Mtb* $\sigma^H$-RPo structure. **a** Recognition of the "$G_{-11}T_{-10}T_{-9}$" sequence in the −10 element by $\sigma^H_2$. **b** Stacking of −7(nt) and −6(nt) nucleotides by RNAP-β subunit and $\sigma^H_2$. **c** The recognition of $G_{-5}$(nt) by RNAP-β subunit and $\sigma^H_2$. **d** The interaction between RNA polymerase (RNAP) and CRE element. Colors are as above. **e** The interaction between RNAP and DNA/RNA hybrid. **f** The in vitro transcription activity of RNAP derivatives comprising alanine substitutions of DNA-contacting residues on RNAP-β subunit and $\sigma^H$. The experiments were repeated in triplicate, and the data are presented as mean ± S.E.M. Source data of **f** are provided as a Source Data file

mutational analyses. Our study demonstrates the structural basis for RNAP holoenzyme formation and transcription initiation by the ECF σ factors.

Our structures show that $\sigma^H$ binds to RNAP in a similar way to $\sigma^A$, in which $\sigma_2$ and $\sigma_4$ stay on the surface of RNAP, and $\sigma_{3.2}$ inserts into the active center. The interactions of $\sigma^H_2$ and $\sigma^H_4$ with RNAP were thus anticipated and supported a structure model of *E. coli* $\sigma^E$-RNAP holoenzyme[45], as the residues contacting the βFTH and β′CC domains are conserved between the ECF and primary σ factors. However, the interactions of $\sigma^H_{3.2}$ with RNAP are unexpected, showing similarity in neither sequence nor secondary structure between the $\sigma_{3.2}$ regions of $\sigma^{ECF}$ and $\sigma^A$ (Supplementary Fig. 3). In vitro transcription experiments show that $\sigma^H_{3.2}$ is essential for the transcription activity of $\sigma^H$-RNAP; removing or replacing the linker with an unrelated sequence completely abolished its transcription activity (Fig. 2d). It is worth noting that the B-reader loop of TFIIB reaches into the active site cleft of yeast pol II in a similar way to $\sigma_{3.2}$[46]. Considering that this general mode of interaction appears to be conserved between prokaryotic RNAP and eukaryotic pol II, it is reasonable to propose that the $\sigma_2/\sigma_4$ linker of other bacterial ECF σ factors very likely also inserts into the active site cleft of RNAP. Our transcription assays show that chimeric ECF σ factors with swapped $\sigma_{3.2}$ domains retain function in transcription, albeit with reduced activity (Fig. 2d and Supplementary Fig. 4e), supporting this idea. An intriguing question to be answered is how RNAP uses the same channels to accommodate different $\sigma_{3.2}$ domains.

$\sigma^A$-RPo crystal structures show that $\sigma^A_{3.2}$ contacts nucleotides at the template strand of ssDNA and pre-organizes it into an A-form helical conformation in a manner compatible with pairing of initial nucleotide triphosphates (NTPs)[24,26]. These interactions provide explanations for the effects of $\sigma^A_{3.2}$ on de novo RNA synthesis[26,47,48]. The structure predicts that the $\sigma^A_{3.2}$ finger has to be displaced by an RNA molecule of >4-nt in length and that the $\sigma^A_{3.2}$ loop in the RNA exit channel has to be cleared during the promoter escape process. The interactions observed in the $\sigma^A$-

RPo structure underscore the key role of the $\sigma^A_{3.2}$ on abortive production, pausing, and promoter escape in transcription initiation[28,29,49,50]. In our crystal structures of $\sigma^H$-RPo, we show that $\sigma^H_{3.2}$ guides the template ssDNA into the active site cleft and forms interactions with template ssDNA (Figs. 3d and 5e). We propose that $\sigma^H_{3.2}$ probably functions similarly to $\sigma^A_{3.2}$ during transcription initiation: by stabilizing the template ssDNA and facilitating binding of initial NTPs. The crystal structure of $\sigma^H$-RPo also indicates that $\sigma^H_{3.2}$ should collide with RNA molecules >7 nt in length and that the $\sigma^H_{3.2}$ domain must dissociate from the RNAP RNA exit channel during promoter escape (Fig. 5e), raising the possibility that the $\sigma_{3.2}$ of the ECF factors functions like $\sigma^A_{3.2}$ during abortive production, pausing, and promoter escape in transcription initiation.

Our structure of $\sigma^H$-RPo suggests that substantial differences exist between how individual domains $\sigma^H$-RNAP and $\sigma^A$-RNAP interact with their cognate promoter DNA. Both $\sigma^H_4$ and $\sigma^A_4$ use the same α-helix to bind the −35 element, but the positions of DNA on the α-helix differ by one α-helical turn, resulting a ~ 4 Å difference in the position of the −35 element on the $\sigma_4$ surface (Supplementary Fig. 6B). A previous crystal structure of the *E. coli* $\sigma^E_4$/−35 element binary complex is superimposable on our $\sigma^H$-RPo (Supplementary Fig. 6b)[33], suggesting that the distinct mode of interaction that we observed with the −35 element is likely used by other ECF σ factors.

$\sigma^H$-RNAP "reads" the sequence of the −10 element differently than does $\sigma^A$-RNAP. In our crystal structure of $\sigma^H$-RPo, we discovered that base moieties of three nucleotides—$T_{-10}$(nt), $T_{-9}$(nt), and $G_{-5}$(nt) (corresponding to the positions (−12), (−11) and (−7) of $\sigma^A$-regulated promoters)—were flipped out and inserted into three respective protein pockets on $\sigma^H$ (Figs. 3d and 5a–c), in contrast to the two protein pockets known for base moieties of $A_{(-11)}$(nt) and $T_{(-7)}$(nt) on $\sigma^{A3,24}$. This extra pocket for $T_{-10}$(nt) on $\sigma^H$ was also suggested in a previous structure of a *E. coli* $\sigma^E_2$/−10 element binary complex (Supplementary Fig. 6g)[34]. Sequence alignment of multiple ECF σ factors and $\sigma^A$ revealed that

residues forming the $T_{-10}$(nt) pocket are generally conserved between ECF σ factors but are distinct from $σ^A$, suggesting that other ECF σ factors likely also recognize the nucleotide at this position using similar protein pockets (Supplementary Fig. 3).

$σ^H$-RNAP uses different protein regions to accommodate the flipped guanine base of $G_{-5}$(nt) (corresponding to position (−7) of $σ^A$-regulated promoters) than does $σ^A$-RNAP for $T_{(-7)}$(nt) (Supplementary Fig. 6i–k). The guanine base of $G_{-5}$(nt) is sandwiched between $σ^H_2$ and the RNAP-β gate loop, while the thymine base of $T_{(-7)}$(nt) resides in a pocket on $σ^A_{1.2}$[3,24]. Our mutation study of the $G_{-5}$(nt) pocket residues demonstrated that the RNAP-β gate loop functions to recognize this particular nucleotide (Fig. 5f), thus raising the possibility that other $σ^{ECF}$-RNAP holoenzymes may also bind and read a nucleotide in the nontemplate ssDNA in a manner analogous to $σ^H$-RNAP.

$σ^H$-RNAP also engages the −10 element differently than does $σ^A$-RNAP. We found that the protein pockets for $T_{-9}$(nt) and $G_{-5}$(nt) on $σ^H$-RNAP do not exist in the absence of promoter DNA (Fig. 6a, b). In the crystal structure of $σ^H$-RNAP, the specificity loop, which recognizes the $T_{-9}$(nt) is disordered; and the RNAP-β gate loop is too far away from the $σ^H_2$ to form the $G_{-5}$(nt) pocket (Fig. 6a, b). Such conformational differences support an "induced-fit" model of interaction between $σ^H$-RNAP and nontemplate ssDNA, in contrast to the accepted "lock-and-key model" for the interaction between $σ^A$-RNAP and nontemplate ssDNA (Fig. 6c, d)[3,24,51].

$σ^H$-RNAP recognizes the $G_{+2}$(nt) of CRE in a same way as does $σ^A$-RNAP (Supplementary Fig. 7d–f). As the residues that form the "G" pocket are solely from the RNAP core enzyme, it is possible that other ECF σ-RNAP holoenzymes are probably able to read the sequence identity of nucleotide at position +2 of the promoter DNA. However, whether the sequence content at this position affects other events (transcription start site selection, slippage synthesis, etc.) during transcription initiation by ECF σ-RNAP as $σ^A$-RNAP remains to be determined[52].

Our crystal structures suggest that $σ^H$ employs a distinct mechanism to unwind promoter DNA compared to $σ^A$ (Supplementary Figure 6C-E): (1) $σ^H$ and $σ^A$ use residues with positions that differ by one α-helical turn on the $σ_{2.3}$ α-helix (N88 for Mtb $σ^H$ vs. W433/W434 for Ec $σ^A$, or W256/W257 for Taq $σ^A$) to unwind promoter DNA; (2) $σ^H$ and $σ^A$ unwind promoter DNA at positions differing by one base pair ((−13)/(−12) junction for $σ^H$ vs. −(12)/(−11) junction for $σ^A$); and (3) $σ^H$ traps and reads two unwound nucleotides ($T_{(-12)}$(nt) and $T_{(-11)}$(nt) immediately after the unwinding points), whereas $σ^A$ only traps and reads one unwound nucleotide ($A_{(-11)}$(nt)). Although it is unclear whether trapping of the flipped nucleotides initiates or facilitates the event of promoter unwinding, such interactions play crucial roles during RPo formation.

Campagne et al. recently identified a similar protein pocket on E. coli $σ^E_2$ for the $T_{(-12)}$(nt) in a crystal structure of E. coli $σ^E$ bound to the −10 element ssDNA, and predicted that E. coli $σ^E$ unwinds promoter dsDNA at the (−13)/(−12) junction[34]. Our crystal structure of $σ^H$-RPo clearly confirms the unwinding position proposed in the study by Campagne et al.. Sequence alignment of multiple ECF σ factors and $σ^A$ revealed that most of the ECF σ factors do not contain the tryptophan dyad of $σ^A$ at corresponding positions, but instead share a conserved (−12) pocket (Supplementary Figure 3). Therefore, it is possible that the ECF σ factors share the same unwinding mechanism as Mtb $σ^H$ and Ec $σ^E$.

Our mutation study of the $σ^H$-regulated promoter showed that substitution of the consensus sequence at almost every position on the −35 element and −10 element abolished transcription activity (Supplementary Figure 5B). Moreover, extending or shortening the spacer of −35/−10 elements substantially reduced

promoter activity (Fig. 2b). These results confirmed previously reported observations that ECF σ factors require a consensus sequence at the −35/−10 elements as well as a rigid spacer on promoter DNA to efficiently initiate transcription[8,32]. Our structures and results from biochemical experiments provide explanations for the promoter stringency of $σ^H$. We show that the interactions among $σ^H$ and RNAP, the unwinding mechanism, and the induced-fit mode of promoter recognition work in concert to collectively confer the high specificity exhibited by $σ^H$ and probably by other ECF σ factors as well.

We have shown that $σ^H$ employs residues different from $σ^A$ to unwind promoter DNA. The well-conserved tryptophan dyad of $σ^A$ functions very efficiently for promoter unwinding; substitutions of the tryptophan dyad in $σ^A$ resulted in severely reduced transcription activity[43,44]; and sequence variations at corresponding positions account for inferior DNA unwinding capacity of other alternative σ factors[25]. Given that ECF σ factors lack the tryptophan dyad at corresponding positions, we infer $σ^H$ (and probably other ECF σ factors) unwinds promoter DNA less efficiently than does $σ^A$. This putative sub-optimal unwinding efficiency could be compensated by employing a very-high-affinity consensus sequence of promoter DNA to facilitate its loading[4,8,25]. Our proposed induced-fit mode of nontemplate ssDNA binding by $σ^H$-RNAP at the position immediate downstream of unwinding—i.e. $T_{-9}$(nt)—also require the consensus promoter sequence to induce formation of correct conformation of the "specificity loop" (Fig. 6c); RNAP is not able to efficiently propagate promoter unwinding downstream without firmly anchoring the "master" nucleotide—$A_{-11}$(nt) for E. coli $σ^{70}$ corresponding to $T_{-9}$(nt) for Mtb $σ^H$—as demonstrated in the case of E. coli $σ^{70}$-RNAP[53–55].

In conclusion, we demonstrate the structural basis of RNAP holoenzyme formation and transcription initiation by the ECF σ factors, thereby deepening our understanding the basic mechanisms of transcription initiation used by the largest and most diverse group of bacterial initiation factors. Our work will facilitate the rational design of orthogonal transcription units based on ECF σ factors and should help computational chemistry and other efforts to design selective antibacterial agents through the inhibition ECF σ factor-mediated transcription initiation.

## Methods

**Plasmid construction.** The plasmids used in this study are listed in Supplementary 1. For construction of the expression plasmid pTolo-EX5-Mtbσ$^H$, the M. tuberculosis $σ^H$ gene amplified from M. tubercolusis genomic DNA (see Supplementary Data 1 for primer information) was cloned into the pTolo-EX5 plasmid (Tolo Biotech.) using NcoI and XhoI restriction sites. The pTolo-EX5-Mtbσ$^H$ derivatives bearing single or double mutations were generated through site-directed mutagenesis (Transgen biotech).

The pTolo-EX5-Mtbσ$^H$ derivatives encoding chimeric $σ^H$ were generated by replacing the DNA fragment encoding Mtb $σ^H_{3.2}$ (aa 96–144) with DNA fragments encoding Ec $σ^A$ (aa 164–212; disordered acidic loop of the non-conserved region), Mtb $σ^E_{3.2}$ (aa 150–189), Mtb $σ^L_{3.2}$ (aa 78–122), or Mtb $σ^M_{3.2}$ (aa 98–137) in pTolo-EX5-Mtbσ$^H$ (Tolo Biotech).

The pACYCDuet-Mtb-rpoA-rpoZ plasmid was constructed by replacing Mtb rpoD with Mtb rpoZ in parent plasmid pACYCDuet-Mtb-rpoA-sigA plasmid using KpnI and NdeI (Supplementary Table 1). The pETduet-Mtb-rpoB-rpoC derivatives bearing single mutations were generated through site-directed mutagenesis (Transgen Biotech; Supplementary Table 1 and Supplementary Data 1).

For construction of plasmids for in vitro transcription assays of Mtb $σ^H$, the promoter region (−50 to +51) of ClpB gene amplified from M. tuberculosis genomic DNA was cloned into pEASY-Blunt simple vectors, resulting in pEASY-Blunt-pClpB (Transgen Biotech; Supplementary Table 1 and Supplementary Data 1). The derivatives of pEASY-Blunt-pClpB with varied −35/−10 spacer lengths were obtained by site-directed mutagenesis (Supplementary Figure 2J). The promoter region (−50 to +51) of Rv2466c gene amplified from M. tuberculosis genomic DNA was cloned into pEASY-Blunt simple vectors, resulting in pEASY-Blunt-pRv2466c (Supplementary Table 1 and Supplementary Data 1; Supplementary Figure 2L).

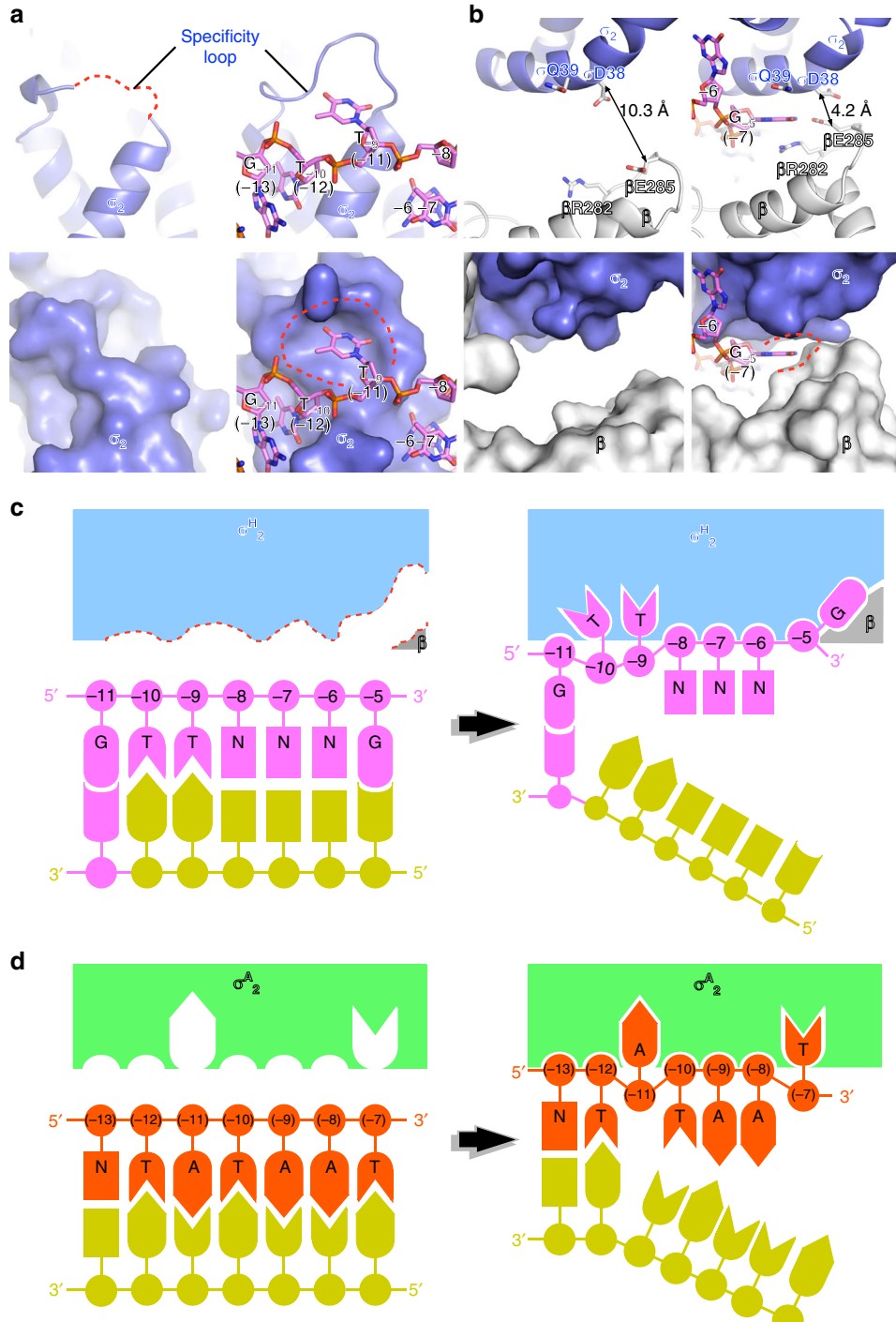

**Fig. 6** The induced-fit mechanism of promoter recognition by *Mtb* σ^H-RNAP. **a** The T_{−9}(nt) pocket does not exist in σ^H-RNAP holoenzyme (left) but is induced by DNA binding in σ^H-RPo (right) structures. **b** The G_{−5}(nt) pocket does not exist in σ^H-RNAP holoenzyme (left) but is induced by DNA binding in σ^H-RPo (right) structures. The pockets are presented in cartoon (top) and surface (bottom). **c** The schematic of induce-fit model of promoter recognition by σ^H-RNAP. **d** The schematic of lock-and-key model of promoter recognition by σ^A-RNAP

The derivatives of pARTaq-N25–100-TR2 for in vitro transcription assays of *Mtb* σ^A with varied −35/−10 spacer lengths were obtained by site-directed mutagenesis (Supplementary Figure 2K).

**Protein preparation**. For preparation of *M. tuberculosis* σ^H, *E. coli* BL21(DE3) cells (NovoProtein) carrying pTolo-EX5-*Mtb*σ^H were cultured in Luria-Bertani broth (LB) at 37 °C, and the expression of N-terminal sumo-tagged *Mtb* σ^H was induced at 18 °C for 14 h with 0.5 mM isopropyl β-D-1-thiogalactopyranoside (IPTG) at OD_{600} of 0.8. Cells were harvested by centrifugation (8000 × *g*, 4 °C), re-

suspended in lysis buffer (20 mM Tris-HCl (pH 8.0), 0.5 M NaCl, 5% (v/v) glycerol, 0.5 mM β-mercaptoethanol, and protease inhibitor cocktail (bimake.cn)) and lysed using an Avestin EmulsiFlex-C3 cell disrupter (Avestin, Inc.). The lysate was centrifuged (16,000 × *g*; 45 min, 4 °C) and the supernatant was loaded on to a 2 mL column packed with Ni-NTA agarose (SMART, Inc.). The protein was washed by lysis buffer containing 20 mM imidazole and eluted with lysis buffer containing 250 mM imidazole. The eluted fraction was digested by tobacco etch virus protease and dialyzed overnight in dialysis buffer (20 mM Tris-HCl (pH 8.0), 0.2 M NaCl, 1% (v/v) glycerol, and 0.5 mM β-mercaptoethanol). The sample was loaded onto a second Ni-NTA column and the cleaved protein was retrieved from the flow-

through fraction. The sample was diluted to the dialysis buffer with 0.05 M NaCl and further purified through a Heparin column (HiTrap Heparin HP 5 mL column, GE Healthcare Life Sciences) with buffer A (20 mM Tris-HCl (pH 8.0), 0.05 M NaCl, 1% (v/v) glycerol, and 1 mM dithiothreitol (DTT)) and buffer B (20 mM Tris-HCl (pH 8.0), 1 M NaCl, 1% (v/v) glycerol, and 1 mM DTT). Fractions containing *M. tuberculosis* $\sigma^H$ was concentrated to 5 mg/mL and stored at −80 °C. The *M. tuberculosis* $\sigma^H$ derivatives were prepared by the same procedure.

For preparation of selenomethionines (SeMet)-labeled *M. tuberculosis* $\sigma^H$, BL21 (DE3) strains carrying pTolo-EX5-*Mtb*$\sigma^H$ were cultured in SelenoMet base medium supplemented with nutrient mix (Molecular Dimensions) at 37 °C. The amino-acid mixture containing selemethionine was added into the culture at $OD_{600}$ of 0.4 and the protein expression was induced with 0.5 mM IPTG at $OD_{600}$ of 0.8 for 14 h at 18 °C. The SeMet-labeled *Mtb* $\sigma^H$ was purified as described above.

The *M. tuberculosis* RNAP core enzyme was expressed and purified from *E. coli* BL21(DE3) carrying pETDuet-*Mtb*-rpoA-rpoZ and pACYCDuet-*Mtb*-rpoB-rpoC as described[56]. The protein sample was concentrated to 5 mg/mL and stored at −80 °C.

**Nucleic acid scaffolds**. Nucleic acid scaffolds for assembly of $\sigma^H$-RPo* for crystallization of $\sigma^H$-RNAP holoenzyme were prepared from synthetic oligos (non-template DNA: 5′-GTTGTGCTGGGCGTCACGGATGCA-3′; template DNA: 5′-TGCATCCGTGAGTCGGT-3′, Sangon Biotech, Supplementary Figure 1A) by an annealing procedure (95 °C, 5 min followed by 2 °C-step cooling to 25 °C) in annealing buffer (5 mM Tris-HCl, pH 8.0, 200 mM NaCl, and 10 mM $MgCl_2$).

Nucleic acid scaffolds for crystallization of $\sigma^H$-RPo were prepared from synthetic oligos (nontemplate DNA: 5′-CGGAACAGTTGCGACTTAGACGTGGTTGTGG GAGCTGCTATACTCTCC-3′; template DNA: 5′-GGAGAGTATAGGTCGAGG GTGTACCACGTCTAAGTCGCAACTGTTCC-3′, Sangon Biotech; and RNA: 5′-CCCUCGA-3′, Genepharma; Fig. 3c) by an annealing procedure (95 °C, 5 min followed by 2 °C-step cooling to 25 °C) in annealing buffer (5 mM Tris-HCl, pH 8.0, 200 mM NaCl, and 10 mM $MgCl_2$).

**M. tuberculosis $\sigma^H$-RPo complex reconstitution**. The *M. tuberculosis* $\sigma^H$-RPo and $\sigma^H$-RPo* were reconstituted from *M. tuberculosis* RNAP core enzyme, $\sigma^H$ (or SeMet-$\sigma^H$), and nucleic acid scaffolds. The RNAP core enzyme, $\sigma^H$, and nucleic acid scaffolds were mixed at a 1:4:1.2 molar ratio and incubate at 4 °C overnight. The mixture was loaded on a HiLoad 16/60 Superdex S200 column (GE Healthcare, Inc) equilibrated in 20 mM Tris-HCl (pH 8.0), 0.1 M NaCl, 1%(v/v) glycerol, and 1 mM DTT. Fractions containing *Mtb* $\sigma^H$-RPo were collected, concentrated to 7.5 mg/mL, and stored at −80 °C.

**Structure determination of M. tuberculosis $\sigma^H$-RNAP holoenzyme**. The structure of $\sigma^H$-RNAP holoenzyme was obtained during an attempt for obtaining the $\sigma^H$-RPo* with the fork transcription bubble DNA scaffold (no RNA oligo in the scaffold). The initial screen was performed by a sitting-drop diffusion technique. Crystals grown from optimized reservoir solution A (1 μL 0.2 M NaAc, 0.1 M sodium citrate (pH 5.5), and 10% PEG4000 mixed with 1 μL 7.5 mg/mL protein complex) for 3 days at 22 °C were harvested for X-ray diffraction data collection. Crystals were soaked in stepwise fashion to reservoir solution A containing 18%(v/v) (2R, 3R)-(−)-2,3-butanediol (Sigma-Aldrich) and cooled in liquid nitrogen. The crystals of $\sigma^H$-RNAP derivative containing SeMet-labeled $\sigma^H$ were obtained by analogous procedure.

Data were collected at Shanghai Synchrotron Radiation Facility (SSRF) beamlines 17U and 19U1, processed using HKL2000[57]. The structure was solved by molecular replacement with Phaser MR[58] using the structure of *M. smegmatis* core enzyme in a *M. smegmatis* transcription initiation complex (PDB: 5TW1)[35] [https://www.rcsb.org/structure/5TW1] as the search model. Only one molecule of RNAP core enzyme was found in one asymmetric unit. The electron density maps show clear signal for $\sigma^H$. Cycles of iterative model building and refinement were performed in Coot[59] and Phenix[60]. Residues of $\sigma^H$ were built into the model at the last stage of refinement. No density of nucleic acid was observed in all stages of refinements, suggesting that the nucleic acids dissociated during crystallization resulting in a crystal of $\sigma^H$-RNAP holoenzyme. The final model of *Mtb* $\sigma^H$-RNAP holoenzyme was refined to $R_{work}$ and $R_{free}$ of 0.218 and 0.258, respectively. Analogous procedures were used to refine the structures of $\sigma^H$-RNAP holoenzyme with SeMet-labeled $\sigma^H$.

**Structure determination of M. tuberculosis $\sigma^H$-RPo**. The initial screen of $\sigma^H$-RPo was performed by a sitting-drop vapor diffusion technique. Crystals grown from reservoir solution B (1 μL 2% Tacsimate pH 5.0, 0.1 M Sodium citrate pH 5.6, 16 % PEG3350 mixed with 1 μL 7.5 mg/mL protein complex) for 15 days at 22 °C were harvested for X-ray diffraction data collection. Crystals were soaked in stepwise fashion to the reservoir solution B containing 18%(v/v) (2R, 3R)-(−)-2,3-butanediol (Sigma-Aldrich) and cooled in liquid nitrogen. Data were collected at SSRF beamlines 17U and 19U1, processed using HKL2000[57]. The structure was solved by molecular replacement with Phaser MR[58] using the structure of *M. tuberculosis* $\sigma^H$-RNAP as a search model. Only one molecule of $\sigma^H$-RNAP was found in one asymmetric unit. The electron density map showed clear signals for nucleotides in transcription bubble and downstream DNA duplex after initial rigid-body refinement, and clear signals for nucleotides in upstream DNA duplex after iterative

cycles of model building and refinements in Coot[59] and Phenix[60]. The nucleotides were built into the model at the last stage, and the final model of *Mtb* $\sigma^H$-RPo was refined to $R_{work}$ and $R_{free}$ of 0.220 and 0.255, respectively.

**In vitro transcription assay**. Transcription assays with *M. tuberculosis* RNAP $\sigma^H$-holoenzyme were performed as follows: reaction mixtures contained (20 μL): 80 nM *M. tuberculosis* RNAP core enzyme; 1 μM *M. tuberculosis* $\sigma^H$; 40 mM Tris-HCl, pH 7.9; 75 mM KCl; 5 mM $MgCl_2$; 2.5 mM DTT; and 12.5% glycerol. Reaction mixtures were incubated for 10 min at 37 °C, and then supplemented with 2 μL promoter DNA (1 μM; amplified from pEASY-Blunt-p*ClpB*; Supplementary Data 1), and further incubated for 10 min at 37 °C. The reaction was initiated by adding 0.7 μL NTP mixture (3 mM [α-$^{32}$P]UTP (0.04 Bq/fmol), 3 mM ATP, 3 mM GTP, and 3 mM CTP), and RNA synthesis was allowed to proceed for 10 min at 37 °C. Reactions were terminated by adding 8 μL loading buffer (10 mM EDTA, 0.02% bromophenol blue, 0.02% xylene cyanol, and 98% formamide), boiled for 2 min, and stored in ice for 5 min. Reaction mixtures were applied to 15% urea-polyacrylamide slab gels (19:1 acrylamide/bisacrylamide), electrophoresed in 90 mM Tris-borate (pH 8.0) and 0.2 mM EDTA, and analyzed by storage-phosphor scanning (Typhoon; GE Healthcare, Inc.).

Transcription assays with *M. tuberculosis* RNAP $\sigma^A$-holoenzyme were performed essentially as above except that $\sigma^A$ instead of $\sigma^H$ were added and N25 promoter DNA were used (amplified from pARTaq-N25-100-TR2; Supplementary Table 1 and Supplementary Data 1).

Transcription assays using *M. tuberculosis* RNAP $\sigma^H$-holoenzyme and p*Rv2466c* promoters were also performed essentially as above with subtle modifications. The reaction mixtures (20 μL) containing 160 nM *M. tuberculosis* RNAP core enzyme, 1 μM *M. tuberculosis* $\sigma^H$, 40 mM Tris-HCl, pH 7.9, 75 mM KCl, 5 mM $MgCl_2$, 2.5 mM DTT, and 12.5% glycerol were incubated for 10 min at 37 °C, and then supplemented with 2 μL promoter DNA (1 μM, amplified from pEASY-p*Rv2466*c; Supplementary Table 1 and Supplementary Data 1), and further incubated for 10 min at 37 °C. The reactions were initiated by adding 4 μL NTP mixture (0.1 mM ATP, 0.1 mM GTP, 0.1 mM CTP, and 7 μM [α-$^{32}$P]UTP (5.6 Bq/fmol)) and were allowed to proceed for 20 min at 37 °C. The reactions were terminated and the transcripts were separated and visualized as above.

**Quantification and statistical analysis**. All biochemical assays were performed at least three times independently. Data were analyzed with SigmaPlot 10.0 (Systat Software Inc.).

**Reporting summary**. Further information on experimental design is available in the Nature Research Reporting Summary linked to this article.

## Data availability
The accession numbers for the coordinates and structure factors for *M. tuberculosis* $\sigma^H$-RNAP and $\sigma^H$-RPo in this paper are PDB: 5ZX3 and 5ZX2, respectively. The source data underlying Figs. 2b, d, 4c, d, f, and 5f, and Supplementary Figs. 2i, 4e, and 5b are provided as a Source Data file. Other data are available from the corresponding author upon reasonable request.

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

## Acknowledgements

Work was supported by the Strategic Priority Research Program of the Chinese Academy of Sciences (XDB29020000), the National Natural Science Foundation of China (31670067 and 31822001), and the Leading Science Key Research Program of the Chinese Academy of Sciences (QYZDB-SSW-SMC005). We thank Prof. Richard Ebright for generous gifts of pACYCDuet-*Mtb-rpoA-rpoD*, pETDuet-*Mtb-rpoB-rpoC*, and pARTaq-N25–100-TR2 constructs; Prof. Xiaoming Zhang for generous gift of *M. tuberculosis* genomic DNA; and Tolo Biotechnology for generous gift of pTolo-EX vectors. We thank the staff at beamline BL18U1/BL19U1 of National Center for Protein Science Shanghai (NCPSS), and at beamline BL17U1 of Shanghai Synchrotron Radiation Facility for assistance during data collection.

## Author contributions

L.L. solved the structures. C.F. performed in vitro transcription assays. N.Z. assisted in structure determination. T.W. prepared pACYCDuet-*Mtb-rpo*A-*rpoZ* and purified proteins. Y.Z. designed experiments, analyzed data, and wrote the manuscript.

## Additional information

**Competing interests:** The authors declare no competing interests.

