## [Peer Review File · Nature Communications]

Reviewers' comments:

Reviewer #1 (Remarks to the Author):

In this study, the authors report two ~2.8Å-resolution crystal structures of *Mycobacterium tuberculosis* transcription initiation complexes containing the ECF alternative sigma factor sigmaH. In the first crystal structure, RNA polymerase bound to sigmaH was resolved. In the second, a structure was obtained for a complex of RNAP, sigmaH, and promoter nucleic acid scaffold comprising upstream and downstream DNA, a mismatch transcription bubble, and a 7-nt RNA primer. The structures revealed the interactions between sigmaH, RNAP, and promoter DNA and lend structural insight into sigmaH's recognition of promoter -35 and -10 elements as well as of the CRE. Additionally, sigmaH was shown to open one additional base pair on the upstream edge of the transcription bubble as compared with sigmaA. Mutational analysis was applied to demonstrate the functional significance of observed interactions.

I recommend that the manuscript be accepted pending the revisions detailed below.

Comments:

1. Experimental densities should be shown for sigmaH and nucleic acids, either instead of or in addition to refined 2Fo-Fc maps. Figure S1A: experimental density for sigmaH within the holoenzyme (before any modeling of the sigma factor) should be included. Figure 3D (or possibly added to Fig. S4): experimental density for nucleic acids (before any were modeled)

2. The structure of the RNAP-sigmaH holoenzyme revealed an additional region of RNAP, the betaCTH, which serves as an anchor point for sigmaH4 and may restrain its movability. The authors propose that this anchor point restrains sigmaH4 and the restrained mobility relative to sigmaA4 contributes to sigmaH4's preference for a 17-bp spacer between the -35 and -10 promoter elements.

To test this hypothesis, the authors conduct transcription assays in the presence of wild type or betaCTH-deletion mutant RNAP using promoters with various spacer lengths as templates (Fig. 2B).

The description of these transcription assays is ambiguous. What are the sequences of the pCIPB promoter variants, how long are the DNA templates, and are the downstream sequences also derived from the pCIPB gene? Are these fully double-stranded or bubble templates? Further explanation of the DNA templates used ("In vitro transcription assay" section of Methods) would help clarify.

The experiments shown in Fig. 2B show that upon deletion of the betaCTH, sigmaH is less able to discriminate between spacer lengths, but largely as the result of a defect in transcription of the most favored spacer lengths. Given only the data and literature described in the text, this result on its own does not convincingly indicate that the effect of the betaCTH is necessarily to restrain the movement of sigmaH (p. 6, paragraph 2 and p. 19, paragraph 2) For example, it could be necessary to properly position the domain and/or contribute to the affinity of sigmaH for RNAP. It is also not evident from the information presented here whether a betaCTH deletion has any effect on RNAP itself or sigmaA-dependent transcription. The authors could either provide additional data or references to strengthen their argument, or rewrite this conclusion as more speculative.

3. What is the length of the termination product being quantified in the in vitro transcription reactions (Figs. 2, 4, 5, S5)? In the experiment shown in Fig. 2D, do the linker chimaeras have an effect on both abortive and productive transcription? An analysis of abortive transcripts could potentially be informative, given that the collision between sigmaH3.2 and nascent RNA would occur at a longer nucleotide length (>7nt) than collisions between the nascent RNA and sigmaA (>4 nt) as described on p. 16, first paragraph. Perhaps one representative lane could be included

in the supplementary material.

4. Although a Mtb RNAP-sigmaA structure is available, several comparisons between sigmaH and sigmaA are made across species, and it is unclear from reading the text and figure legends why this was done. I would suggest that when different species are shown in comparison figures, the authors include a brief statement (e.g. in the figure legend) explaining why the Mtb-sigma factor structure wasn't used for comparison:

- (a) Fig 1: comparison to Tth sigmaA
- (b) Fig S6B: comparison to Msm sigmaA
- (c) Fig. S6D comparison to Taq sigmaA

Minor points:

1. Please rephrase the sentence on p. 9 regarding the E. coli sigmaE4 and S. coelicolor sigmaR4-dsDNA structures, since it sounds like a structure of the co-complex of sigmaR4 and dsDNA was solved rather than modeled based on the apo-sigmaR4 structure.

2. Please rephrase the sentence "chimeric ECF sigma factors with swapped sigma3.2 domains function normally (Figure 2D)" (p. 15, second paragraph). Although the chimeric factors still function in transcription, the levels of activity are markedly reduced.

3. Please change "detail interactions" to "detailed interactions" in the supplementary material figure titles/legends and correct the typo on p. 18, second heading "promoter recognition".

Reviewer #2 (Remarks to the Author):

This manuscript presents two crystal structures at ~ 2.8 Å resolution: the M. tuberculosis RNA polymerase-ECF holoenzyme and the initial transcribing complex using a promoter DNA with pre-opened transcription bubble and a 7 nt synthetic complementary RNA. ECF (extra-cytoplasmic function) sigma factors are alternative sigma factors with distinct functions and mechanisms. A large amount of structural information is available for the major house keeping sigmaA (sigma70), the major alternative sigmaN (sigma54), stress response sigmaS. The structures here present first structures of the holoenzyme and initial transcribing complexes containing an ECF, thus filling in a gap in our knowledge in bacterial transcription. However, the manuscript as it stands, does not provide convincing evidences/arguments to explain the distinct functions/mechanisms of ECF.

Some general and specific comments are listed below:

The introduction is very brief and literature is not referenced sufficiently. It should include information on other sigma factors and how sigma factors are classified into different classes and groups and what the main differences between ECF and other sigma classes lie. The introduction should also briefly summarize what is known about holoenzymes and promoter melting involving other sigma classes (not just sigmaA). The introduction should then summarize what is known about ECF. For example, the key publication by Campagne et al. 2014 should be introduced here, especially what was proposed in that study in terms of promoter melting.

The quality of the proteins and complexes should be shown in supplementary data.

The manuscript should focus on unique features of ECF (compare with sigA, sigS and sigN for example) and how these features explain its functions/mechanisms such as stringent requirement for sequences in -35 and -10 regions.

The authors use the term RPo in describing their DNA complex. The structure actually contains a 7nt RNA, therefore it represents the initial transcribing complex.

The B factors (~ 90 Å²) of the holoenzyme model are high for the resolution of the structure (2.75 Å). Please explain. This could indicate that the resolution is worse than reported. It is important to show electron density for regions that involve DNA interactions to support the specific

interactions described in Figure 4, 5 and 6.

Figures and Supplementary figure numbers are not in the order of being cited in the text. For example, Figure 3C is cited before Figure 3A-B. Figure S3 and S5 cited before Figure S2 and S4. Please check carefully throughout the manuscript.

Relevant references are missing (see points above about Introduction) or references and figures are not cited correctly. For example, the authors state that "Clamp closure as an obligatory step in RPo formation" and cited Figure S4A, which shows comparisons between sigmaH and sigmaA!. Please check thoroughly and cite appropriate references and figures to support the statement/conclusions.

The authors observe that sigmaH region 4 makes additional interactions with β CTH in their holoenzyme structure and propose that this might explain the stringent requirement for spacing between -35 and -10, which they then tested using mutagenesis and in vitro transcription assays (Figure 2). However, it is unclear if this additional interaction is maintained in the promoter-bound complex. What is β CTH involved in other holoenzymes? Data presented in Figure 2B show that upon β CTH deletion, the transcription activity is down by 50% and the differences in activities between different spacers 16-19 are not dramatically different from those observed in WT. These experiments should be compared with those of sigmaA to confirm if β CTH indeed plays a role in ECF regulated transcription (but not in sigmaA regulated transcription).

The authors state that actual melting starts at -10 as shown in the structure although the synthetic oligos at this position are base-paired. However, since the oligos have mis-matched bases from -9 and thus -10 nucleotides already lack the adjacent base stacking, required for stable interactions and structural stability. To really show that DNA starts to melt at -10, the authors need to use other oligos to ensure that the base separation at -10 is not due to structural instability introduced by the lack of base stacking at -9.

Regarding DNA melting, the authors should make it clear that residues involved in ssDNA interactions are not necessarily directly responsible for melting but could be simply used to stabilise the melted out ssDNA as promoter DNA is dynamic and can temporarily spontaneously melt out.

The figures that describe detailed interactions (Figures 3-6) need to be improved. For example, Figures 3A and B are redundant. It might be useful to include comparisons with RNAP-sigmaA initial transcribing complex instead. Figures (4-6) describe detailed interactions and should be related to Figure 3, perhaps by boxing Figure 3.

Figures 4-6, color schemes need to be improved, especially between DNA and proteins, to improve clarity.

Figure 4D, the data are not described properly and do not convincingly support the conclusion that "M181 is the determinant for specificity of positions -30 and -31 of the -35 element"

There are numerous statements/descriptions in the manuscript that are confusing or not supported by data/literature. The authors should go through their text carefully to ensure each statement/conclusion is supported. For example:

The description of the residues that are involved in stabilising upstream transcription bubble is confusing. R84, I85 and R99 for example, are shown in figures but not mentioned in the text. The authors should describe this part in relation to the results published in Campagne et al. 2014.

The authors state that "Our crystal structure shows that the N92 residue of σ H makes a H-bond with the base moiety of C-11(t) of the -11 G:C pair, providing a structural explanation for sequence recognition at this position (Figure 5A)."

However, mutating N92 to A only has modest effects on transcription (Figure 5F). Please clarify and explain.

The authors state "the positions of DNA on the α -helix differ by one α -helical turn, resulting a ~ 4 Å difference in the position of the -35 element on the σ 4 surface (Figure S6B)." The differences are small (not one helical turn). The authors should compare the initial transcribing complex structures directly by aligning on RNAP in order to compare the differences in sigma position and DNA path.

The statement "RNAP is not able to efficiently propagate promoter unwinding downstream without firmly anchoring the thymine base of T-9(nt) in the induced pocket." Please show data or cite appropriate literature.

sigmaH uses N88 to stabilize the strand separation at upstream transcription bubble which is also proposed for sigmaE. Indeed sigmaE also has a N at this position but N is not conserved in a number of other sigma factors. Can the authors speculate the properties required to separate the strands at this unique position?

The "induced fit" model at -11/-10 and -5 positions (Figure 6) is over-stated. At best, data presented here (and in Campagne et al. 2014) suggest that the loop is flexible and only becomes ordered upon stably binding to ssDNA.

Response to referees:

Referee 1, comment 1:

Experimental densities should be shown for σ^H and nucleic acids, either instead of or in addition to refined 2Fo-Fc maps. Figure S1A: experimental density for σ^H within the holoenzyme (before any modeling of the sigma factor) should be included. Figure 3D (or possibly added to Fig. S4): experimental density for nucleic acids (before any were modeled)

Reply:

We have replaced the 2Fo-Fc map with simulated Fo-Fc map of σ^H in Fig. S2A and included simulated Fo-Fc maps of nucleic acids in Fig. 3E, Fig. S6 and S7 in the revised manuscript.

Referee 1, comment 2:

The structure of the RNAP- σ^H holoenzyme revealed an additional region of RNAP, the β CTH, which serves as an anchor point for σ^{H_4} and may restrain its movability. The authors propose that this anchor point restrains σ^{H_4} and the restrained mobility relative to σ^{A_4} contributes to σ^{H_4} 's preference for a 17-bp spacer between the -35 and -10 promoter elements. To test this hypothesis, the authors conduct transcription assays in the presence of wild type or β CTH-deletion mutant RNAP using promoters with various spacer lengths as templates (Fig. 2B). The description of these transcription assays is ambiguous. What are the sequences of the p*ClpB* promoter variants, how long are the DNA templates, and are the downstream sequences also derived from the p*ClpB* gene? Are these fully double-stranded or bubble templates? Further explanation of the DNA templates used ("In vitro transcription assay" section of Methods) would help clarify.

Reply:

We have added the following descriptions in the 'supplemental materials and methods' and a supplemental figure panel (Fig. S2J) to describes sequence of the promoters.

"For construction of plasmids for *in vitro* transcription assays of *Mtb* σ^H , the promoter region (-50 to +51) of *ClpB* gene amplified from *M. tuberculosis* genomic DNA (forward primer: 5'-TAAAATTGAGCGGAACAGAC -3'; reverse primer: 5'-AAATAAAAAGGCCTGCGATTACCAGCAGGCCCGGGTTAAACGAGTCCACGA -3'; the underlined region indicates a sequence of tR2 terminator) was cloned into pEASY-Blunt simple vectors, resulting in pEASY-Blunt-p*ClpB* (Transgen biotech). The derivatives of pEASY-Blunt-p*ClpB* with varied -35/-10 spacer lengths were obtained by site-directed mutagenesis (Fig. S2J)".

Referee 1, comment 3:

The experiments shown in Fig. 2B show that upon deletion of the β CTH, σ^H is less able to discriminate between spacer lengths, but largely as the result of a defect in transcription of the most favored spacer lengths. Given only the data and literature described in the text, this result on its own does not convincingly indicate that the effect of the β CTH is necessarily to restrain the movement of σ^H (p. 6, paragraph 2 and p. 19, paragraph 2) For example, it could be necessary to properly position the domain and/or contribute to the affinity of σ^H for RNAP. It is also not evident from the information presented here whether a β CTH deletion has any effect on RNAP itself or σ^A -dependent transcription. The authors could either provide additional data or references to strengthen their argument, or rewrite this conclusion as more speculative.

Reply:

Thanks for the comment. The second reviewer also pointed out that "the transcription activity is down by 50% of σ^H -RNAP($\Delta\beta$ CTH) but the differences in activities between different spacers 16-19 are not dramatically different from those observed in WT". We agree with reviewers that the current data are not strong enough to state that $\Delta\beta$ CTH is responsible for the -35/-10 spacer preference of σ^H and also to support that β CTH restrains the movement of σ^{H_4} . We rewrote our conclusion in page 6 and page 18 in the revised manuscript. Moreover,

we are also curious how β CTH behaves in σ^A -dependent transcription and intriguingly we found deletion of β CTH causes overall increase of σ^A -dependent transcription.

The revised paragraph in Page 7:

“To explore the contribution of such interaction to the transcription activity of σ^H -RNAP, we performed *in vitro* transcription experiments using wild-type or β CTH-deleted *Mtb* σ^H -RNAP holoenzyme and *pClpB* promoter variants with -35/-10 spacer lengths ranging from 15-19 bp. The wild-type σ^H -RNAP was most transcriptionally active with a promoter of 17-bp spacer (Figure 2B), consistent with a study reporting that most σ^H -regulated promoters have a 17-bp spacer³⁵. The β CTH-deletion variant caused impaired transcription activity from promoter with the optimal spacer length (17 bp) but showed little effect on promoter with sub-optimal spacer lengths (16 bp and 18 bp) (Figure 2B), suggesting that the interactions between β CTH and σ^H are important for the transcription activity of σ^H . Intriguingly, deletion of β CTH caused a general increase of σ^A -dependent transcription activity from promoter with spacer lengths 15-19 bp (Figure S2I).

We also deleted the following sentence in Page 18

“We showed that the CTH and FTH motifs of the RNAP- β subunit together restrain the movement of the σ^H_4 domain and consequently restrains the σ^H -RNAP holoenzyme from binding promoters of varied spacer lengths (Figure 2B)”

Referee 1, comment 4:

What is the length of the termination product being quantified in the *in vitro* transcription reactions (Figs. 2, 4, 5, S5)? In the experiment shown in Fig. 2D, do the linker chimaeras have an effect on both abortive and productive transcription? An analysis of abortive transcripts could potentially be informative, given that the collision between sigmaH3.2 and nascent RNA would occur at a longer nucleotide length (>7nt) than collisions between the nascent RNA and sigmaA (>4 nt) as described on p. 16, first paragraph. Perhaps one representative lane could be included in the supplementary material.

Reply:

The length of termination and run-off RNA product are 82 nt and 122 nt, respectively, which showed essentially the same pattern. The representative run-off products were presented and quantified in the figures. We apologize that we mislabeled as “termination product” in the figures in the manuscript. We have corrected the typos and add the “122 nt” into corresponding figures. The *pClpB* promoter used in Fig. 2D doesn't encode “T” in the first five positions of initial transcribed region and therefore produce no visible abortive transcripts of < 5 nt using [α -³²P]UTP. We performed the same experiment as in Fig. 2D using another σ^H -regulated promoter *pRv2466c* (Fig. S2L). The results in Fig. S4E showed that the linker chimeras have similar effect on both abortive and productive transcription.

Referee 1, comment 5:

Although a *Mtb* RNAP- σ^A structure is available, several comparisons between σ^H and σ^A are made across species, and it is unclear from reading the text and figure legends why this was done. I would suggest that when different species are shown in comparison figures, the authors include a brief statement (e.g. in the figure legend) explaining why the *Mtb*-sigma factor structure wasn't used for comparison:

- (a) Fig 1: comparison to *Tth* σ^A
- (b) Fig S6B: comparison to *Msm* σ^A
- (c) Fig. S6D comparison to *Taq* σ^A

Reply:

We have included the below statement in the figure legends of Fig. 1 and Fig. S6.

- (a) In Fig 1D, “ σ^A in the crystal structure of *T. thermophiles* σ^A -RNAP (PDB: 1IW7) was used for comparison due to no available structure of *Mtb* σ^A -RNAP holoenzyme.”
- (b) In Fig S6, “The *Mtb* σ^A -RPO structure was chosen for superimposition with *Mtb* σ^H -RPO unless the interactions to be compared is unavailable in *Mtb* σ^A -RPO”

Referee 1, comment 6:

Minor points:

1. Please rephrase the sentence on p. 9 regarding the *E. coli* σ^E and *S. coelicolor* σ^R_4 structures, since it sounds like a structure of the co-complex of σ^R_4 and dsDNA was solved rather than modeled based on the apo- σ^R_4 structure.

Reply:

We have rephrased the sentence as follows, “Previous crystal structure of *E. coli* $\sigma^E_{4/-35}$ dsDNA and a structural model of *S. coelicolor* $\sigma^R_{4/-35}$ dsDNA reported a local DNA shape readout (straight helix with a narrow minor groove) at this region” in Page 10.

2. Please rephrase the sentence “chimeric ECF sigma factors with swapped sigma3.2 domains function normally (Figure 2D)” (p. 15, second paragraph). Although the chimeric factors still function in transcription, the levels of activity are markedly reduced.

Reply:

We have rephrased the sentence as follows, “chimeric ECF σ factors with swapped $\sigma_{3.2}$ domains retain function in transcription, albeit with reduced activity” in Page 15.

3. Please change “detail interactions” to “detailed interactions” in the supplementary material figure titles/legends and correct the typo on p. 18, second heading “promoter recognition”.

Reply:

We have corrected the typos.

Referee 2, comment 1:

The introduction is very brief and literature is not referenced sufficiently. It should include information on other sigma factors and how sigma factors are classified into different classes and groups and what the main differences between ECF and other sigma classes lie. The introduction should also briefly summarize what is known about holoenzymes and promoter melting involving other sigma classes (not just σ^A). The introduction should then summarize what is known about ECF. For example, the key publication by Campagne et al. 2014 should be introduced here, especially what was proposed in that study in terms of promoter melting.

Reply:

We have included additional the following sentences and citations in the introduction section as suggested,

In Page 3,

“Bacterial σ factors are classified into two types— σ^{70} - and σ^{54} -type factors based on their distinct structures and mechanisms. The σ^{70} -type factors can be further classified into four groups according to numbers of conserved domains⁴. Group-1 or primary σ factors contain domains $\sigma_{1.1}$, $\sigma_{1.2}$, σ_{NCR} , σ_2 , $\sigma_{3.1}$, $\sigma_{3.2}$, σ_4 ; group-2 σ factors contain all domains except $\sigma_{1.1}$; group-3 σ factors contain σ_2 , $\sigma_{3.1}$, $\sigma_{3.2}$, σ_4 ; while group-4 or ECF σ factors only contain σ_2 and σ_4 ”

In Page 4,

“The group-2 σ factors use the same set of residues to unwind promoter DNA; but the melting residues of group-3 σ factors are not conserved”

In Page 5

“A recent crystal structure of *E. coli* $\sigma^E_{2/-10}$ ssDNA binary complex suggests that bacterial ECF σ factors probably recognize and unwind promoters through a unique mechanism. Specifically, *E. coli* σ^E employs a flexible ‘specificity loop’ to recognize a flipped master

nucleotide of the -10 element and probably unwinds at a distinct position compared with that of σ^{70} by using non-conserved melting residues³³.”

Referee 2, comment 2:

The quality of the proteins and complexes should be shown in supplementary data.

Reply:

We have prepared another supplemental figure (Fig. S1) including the chromatography results on the Supdex S200 size-exclusion column, SDS-PAGE, native-PAGE, and crystal images of the complexes to show the quality of the proteins and complexes in the revised manuscript.

Referee 2, comment 3:

The manuscript should focus on unique features of ECF (compare with σ^A , σ^S and σ^N for example) and how these features explain its functions/mechanisms such as stringent requirement for sequences in -35 and -10 regions.

Reply:

Thanks for the suggestion. As pointed out by the referee, we presented the first structures of bacterial RNAP complexes comprising ECF σ factors. We think it is necessary to make a thorough comparison between ECF σ factors and other types of σ factors to understand their similarity and difference. Therefore, in the manuscript, we described that similarity of interaction mode with RNAP core enzyme. We also described the unique features of ECF σ factors, which probably accounts for stringent promoter recognition of ECF σ factors.

The unique features we have discovered for the σ^H compared with other type of σ factors in the manuscript, include: 1) the unique interaction with RNAP- β CTH with the σ^H_4 domain; 2) the unique mechanism for promoter unwinding (induced-fit *vs.* lock-and-key); 3) the unique interaction of σ_4 with -35 dsDNA; and 4) the unique protein pocket for accommodating the unwound T₋₁₀(nt) and T₋₉(nt) nucleotides. Part of the last two points have been discussed in previous studies (Lane *et al.*, 2006, PMID: 16903784; Campagne *et al.* 2014, PMID: 24531660); we described and discussed in detail of the first two points mainly in the manuscript and we believe that the interactions among σ^H and RNAP, the unique unwinding mechanism, and the induced-fit mode of promoter recognition work in concert confer the high specificity exhibited by σ^H and probably by other ECF σ factors as well. Please see the last sub-section of in the ‘Discussion’.

Referee 2, comment 4:

The authors use the term RPo in describing their DNA complex. The structure actually contains a 7nt RNA, therefore it represents the initial transcribing complex.

Reply:

Thanks for the suggestion. Although the complex comprises a 7-nt RNA, it doesn't have the key characteristic of RP_{itc} (*i.e.* scrunching). Therefore, we think it is better to name our complex as RPo with a 7-nt RNA primer to avoid confusion and to be consistent with the other bacterial RPo structures with a RNA primer (Zhang *et al.*, 2012, PMID: 27284196; Bae *et al.*, 2015, PMID:26349032)

Referee 2, comment 5:

The B factors ($\sim 90 \text{ \AA}^2$) of the holoenzyme model are high for the resolution of the structure (2.75 \AA). Please explain. This could indicate that the resolution is worse than reported. It is important to show electron density for regions that involve DNA interactions to support the specific interactions described in Figure 4, 5 and 6.

Reply:

Thanks for the suggestions. The higher B-factors of σ^H -RNAP holoenzyme compared with σ^H -RPo may reflex the intrinsic conformational mobility of RNAP clamp domains, as demonstrated in Chakraborty *et al.*, 2012 (PMID: 22859489). The resolution of σ^H -RNAP

structure was cut following the same criteria as σ^H -RPO structure ($I/\sigma > 1$ in the highest resolution shell). The electron density map ambiguously shows good densities for the most side-chains of σ^H , allowing us to build a reliable atomic model of σ^H .

We have included simulated Fo-Fc difference maps for nucleic acids of interest in Fig. S6 and Fig. S7 in the revised manuscript.

Referee 2, comment 6:

Figures and Supplementary figure numbers are not in the order of being cited in the text. For example, Figure 3C is cited before Figure 3A-B. Figure S3 and S5 cited before Figure S2 and S4. Please check carefully throughout the manuscript.

Reply:

Thanks for the comment. We have deleted the redundant sentence “The interactions formed by the DNA/RNA hybrid and downstream DNA duplex with RNAP are essentially the same in both σ^H -RPO and σ^A -RPO (Figure 3C)” in Page 9.

We have changed the order of Fig. S2 and Fig. S3 and updated the citations of the two figures in the text. We have removed the unnecessary citations of Fig. S5 in the text. The revised manuscript has proper order of figure citation.

Referee 2, comment 7:

Relevant references are missing (see points above about Introduction) or references and figures are not cited correctly. For example, the authors state that “Clamp closure as an obligatory step in RPO formation” and cited Figure S4A, which shows comparisons between σ^H and σ^A .

Reply:

Thanks for the comment. We intended to claim that the clamp closure is an obligatory step in RPO formation for both group-1 σ and group-4 σ s. We have rephrased the sentence as follows, “supporting the idea that clamp closure is also an obligatory step of RPO formation in ECF σ -mediated transcription initiation” in Page 8.

Referee 2, comment 8:

Please check thoroughly and cite appropriate references and figures to support the statement/conclusions. The authors observe that σ^{H_4} makes additional interactions with β CTH in their holoenzyme structure and propose that this might explain the stringent requirement for spacing between -35 and -10, which they then tested using mutagenesis and in vitro transcription assays (Figure 2). However, it is unclear if this additional interaction is maintained in the promoter-bound complex. What is β CTH involved in other holoenzymes?

Reply:

The additional interaction between β CTH and σ^{H_4} is maintained in the σ^{H_4} -RPO structure. We have added one sentence “In the σ^H -RPO structure, the σ^H makes the same interactions with RNAP as in the structure of σ^H -RNAP holoenzyme” in Page 8 of the revised manuscript.

We have surveyed all available bacterial RNA polymerase structures and found the C-terminal part is disordered in all of the structures. We have summarized it in previous manuscript “The interaction with β CTH was not observed in any of the previously reported bacterial σ^A -RNAP structures.” in Page 7 of the revised manuscript.

Referee 2, comment 9:

Data presented in Figure 2B show that upon β CTH deletion, the transcription activity is down by 50% and the differences in activities between different spacers 16-19 are not dramatically different from those observed in WT. These experiments should be compared with those of σ^A to confirm if β CTH indeed plays a role in ECF regulated transcription (but not in σ^H regulated transcription).

Reply:

Thanks for the constructive comment. The first referee also asked the same question. We have performed the requested experiments with σ^A -RNAP holoenzyme. We carefully revised our conclusion. Please refer to reply to Referee 1, comment 3.

Referee 2, comment 10:

The authors state that actual melting starts at -10 as shown in the structure although the synthetic oligos at this position are base-paired. However, since the oligos have mis-matched bases from -9 and thus -10 nucleotides already lack the adjacent base stacking, required for stable interactions and structural stability. To really show that DNA starts to melt at -10, the authors need to use other oligos to ensure that the base separation at -10 is not due to structural instability introduced by the lack of base stacking at -9.

Reply:

Thanks for the comment. We understand the concern that unpairing of -10 nucleotide could be a result of mismatched -9 nucleotides. However, such design of mismatched nucleotides in the transcription bubble has been widely applied for determining structures of initiation complexes and elongation complexes. In these studies, the base-pairing is not affected by its immediate nearby mismatched nucleotides (Zhang *et al.*, 2012, PMID: 27284196; Bae, *et al.*, 2015, PMID: 26349032; Vassylyev *et al.*, 2007; PMID:17581590). Moreover, we have observed in the structure that both unwound T₋₁₀(nt) and T₋₉(nt) of the non-template ssDNA and the unwound A₋₁₀(t) and T₋₉(t) of the template ssDNA have been firmly secured in specific protein pockets (Fig. 4E) rather than loosely unpaired. More importantly, the base of the unwound T₋₁₀(nt) and T₋₉(nt) of the non-template ssDNA makes base-specific interactions in the pocket. All the evidences suggest that σ^H -RPO unwinds the promoter DNA at the -11/-10 junction.

Referee 2, comment 11:

Regarding DNA melting, the authors should make it clear that residues involved in ssDNA interactions are not necessarily directly responsible for melting but could be simply used to stabilize the melted-out ssDNA as promoter DNA is dynamic and can temporarily spontaneously melt out.

Reply:

Thanks for the suggestion. We have included the following sentence in the revised manuscript, “Although it is unclear whether trapping of the flipped nucleotides initiates or facilitates the event of promoter unwinding, such interactions play crucial roles during RPO formation” in Page 18.

Referee 2, comment 12:

The figures that describe detailed interactions (Figures 3-6) need to be improved. For example, Figures 3A and B are redundant. It might be useful to include comparisons with RNAP-sigmaA initial transcribing complex instead. Figures (4-6) describe detailed interactions and should be related to Figure 3, perhaps by boxing Figure 3. Figures 4-6, color schemes need to be improved, especially between DNA and proteins, to improve clarity.

Reply:

Thanks for the suggestion. We have removed the Fig. 3B, and moved the two panels (comparison between σ^A -RPO and σ^H -RPO) of the Fig. S4 into Fig. 3. We have changed the color for carbon atoms in Fig. 4E, 5A, 5B, 5C, and 6B to make the figures clearer. We have also added a few boxes on figure 3D and descriptions in the corresponding figure legend to relate panels of the following figures.

In Page. 25,

“Red box, interactions with the -35 element (details in Fig. 4A); gray box, interactions with the -35/-10 spacer (details in Fig. 4B); blue box, interactions of the ssDNA in transcription bubble (details in Fig. 4E, 5A-5D); green box, interactions with the DNA/RNA hybrid (details in Fig. 5E)”

Referee 2, comment 13

Figure 4D, the data are not described properly and do not convincingly support the conclusion that “M181 is the determinant for specificity of positions -30 and -31 of the -35 element”

Reply:

Thanks for the comment. We have replaced the sentence with “suggesting that M181 partially accounts for sequence specificity of the two positions.” in Page. 10 of the revised manuscript.

Referee 2, comment 14

There are numerous statements/descriptions in the manuscript that are confusing or not supported by data/literature. The authors should go through their text carefully to ensure each statement/conclusion is supported. For example: The description of the residues that are involved in stabilizing upstream transcription bubble is confusing. R84, I85 and R99 for example, are shown in figures but not mentioned in the text. The authors should describe this part in relation to the results published in Campagne et al. 2014.

Reply:

We have mentioned the R84, I85 in previous manuscript “but the residues at the corresponding positions of \$\sigma^H\$ (R84/I85) are not conserved (Figure S3). Mutating R84 and I85 in \$\sigma^H\$ to tryptophan (I85W, R84W, or I85W/R84W) resulted in substantial loss of transcription activity, confirming that \$\sigma^H\$ opens promoter through a different mechanism than \$\sigma^A\$.”

We have mentioned the R99 in previous manuscript

“K96 and R99 of \$\sigma^{H_2}\$ contact the nucleotide at position -14 (Figure 4B). These interactions probably stabilize the conformation of the upstream DNA duplex, likely promoting the engagement of the upstream duplex with \$\sigma^{H_4}\$ and \$\sigma^{H_2}\$ for subsequent promoter unwinding. Mutating K96 and R99 causes a mild loss of transcription activity, suggesting the importance of these interactions (Figure 4F).” in Page 10 of the revised manuscript.

We have added the sentence “and supporting the unique mechanism proposed for *E. coli* \$\sigma^E\$ ³²” and a citation to Campagne *et al.*, 2014 in Page. 12 of the revised manuscript.

Referee 2, comment 15

The authors state that “Our crystal structure shows that the N92 residue of σ^H makes a H-bond with the base moiety of C-11(t) of the -11 G:C pair, providing a structural explanation for sequence recognition at this position (Figure 5A).” However, mutating N92 to A only has modest effects on transcription (Figure 5F). Please clarify and explain.

Reply to comment:

Thanks for pointing out the issue. After carefully revisiting the structure, we think the I91 probably play more important role in recognizing the -11 G:C pair. We have revised the manuscript as below

“Our crystal structure shows that the base moiety of C-11(t) of the -11 G:C pair makes one H-bond with N92 and extensive Van der Waal interactions with I91, alanine substitution of N92 or I91 resulted in modest or substantial decrease of transcription, respectively (Figure 5A), providing a structural explanation for sequence recognition at this position.”

Referee 2, comment 16

The authors state “the positions of DNA on the α -helix differ by one α -helical turn, resulting a ~ 4 Å difference in the position of the -35 element on the σ^4 surface (Figure S6B).” The differences are small (not one helical turn). The authors should compare the initial transcribing complex structures directly by aligning on RNAP in order to compare the differences in sigma position and DNA path.

Reply:

We did compare the σ^A - and σ^H -RPO structures directly by aligning on RNAP (please see Fig. S6E). The Fig. S6C-E clearly show that σ^A and σ^H use residues at different position on the helix to unwind promoter DNA; σ^A uses W256/W257 (corresponding to R84/I85 of σ^H) while σ^H uses N88. The four-residue difference causes one α -helical-turn shift.

Referee 2, comment 17

The statement “RNAP is not able to efficiently propagate promoter unwinding downstream without firmly anchoring the thymine base of T₋₉(nt) in the induced pocket.” Please show data or cite appropriate literature.

Reply:

Previous reports have shown that either mutating the A₋₁₁(NT) of *E. coli* σ^{70} promoter (Keyduk *et al.*, 2006, PMID: 16531399; Lim *et al.*, 2001, PMID: 11734629) or alanine substitutions of residues forming A₋₁₁(NT) pockets of *E. coli* σ^{70} (Tomsic *et al.*, 2001, PMID: 11443133) resulted in substantially reduced efficiency of RPO formation. Therefore, we proposed that properly anchoring the thymine base of T₋₉(nt) by *Mtb* σ^H is probably also important for RPO formation. Nevertheless, we revised our text to make it clearer,

“RNAP is not able to efficiently propagate promoter unwinding downstream without firmly anchoring the ‘master’ nucleotide-- A₋₁₁(nt) for *E. coli* σ^{70} corresponding to T₋₉(nt) for *Mtb* σ^H --as demonstrated in the case of *E. coli* σ^{70} -RNAP” in Page 19 of the revised manuscript.

Referee 2, comment 18

Sigma H uses N88 to stabilize the strand separation at upstream transcription bubble which is also proposed for sigma E. Indeed, sigma E also has an N at this position but N is not conserved in a number of other sigma factors. Can the authors speculate the properties required to separate the strands at this unique position?

Reply:

As pointed out by the referee, the sequence alignment shows that N88 is the most frequent residue but other polar residues are also present at the corresponding position (H for σ^C and σ^D , and R for σ^I , σ^J , σ^K). It seems that a polar residue with moderate bulky side chain is preferred at this position. It is worthy noting that N88 not only separates the strands, but also makes base-specific interaction with the first unwound nucleotide--T₋₁₀(nt). It is possible that the bulky property of the side-chain is preferred for strand separation and the polar property of the side-chain is preferred for base-specific polar interactions.

Referee 2, comment 19

The “induced fit” model at -10/-9 and -5 positions (Figure 6) is over-stated. At best, data presented here (and in Campagne *et al.* 2014) suggest that the loop is flexible and only becomes ordered upon stably binding to ssDNA.

Reply:

Thanks for the comment. We think the “induced fit” mode is a proper model to explain our structures. The available structures of σ^H -RNAP (in the absence of DNA) and σ^H -RPO (in the presence of DNA) allow a direct structural comparison, which could reveal any conformational change upon DNA binding. The structure superimposition unambiguously shows that the pockets (T-9 and G-5) were induced upon DNA binding. Since the induced conformational change of enzyme upon substrate binding is the key characteristic of the “induced fit” model, the induced formation of protein pockets on σ^H upon ssDNA binding fits very well to the model. This model is in sharp contrast to “lock-and-key” model for ssDNA binding by σ^A , the A-11 and T-7 pockets of which exist regardless of ssDNA binding.

The “induced fit” model for protein-DNA or RNA interaction has been proposed by Dr. Thomas Record (PMID: 8303294), demonstrated by various pairs of protein crystal structures with or without nucleic acids, and summarized in many reviews (Williamson *et al.*, PMID: 11017187; Dyson *et al.*, 2000, PMID: 15738986; Oldfield *et al.*, PMID: 24606139).

REVIEWERS' COMMENTS:

Reviewer #1 (Remarks to the Author):

The authors have adequately addressed my comments, except for one that requires some further clarification.

Regarding the response to comment 1 (experimental density maps): Is the "simulated Fo-Fc difference map" described in Figure S2A a simulated annealing omit difference density? If so, the authors should clarify this in the figure legend and explain exactly what was omitted – i.e. was only sigmaH removed, but were nucleic acids still included? A similar clarification is needed for the "simulated Fo-Fc difference map" shown in Figures S6K, S7A/D, and 3E. If these are composite omit maps, please instead include that information.

After these changes are made, I recommend that the article be accepted for publication.

Reviewer #2 (Remarks to the Author):

The revised manuscript by Li et al. has addressed most of my concerns and I am happy for it to be accepted provided the following points are addressed/clarified.

Figure 2 and associated text on β CTH. The author needs to provide residue numbers for β CTH (both for Mtb and for EColi numbering). The authors stated that "the interaction with β CTH was not observed in any of the previously reported bacterial sigmaA-RNAP structures". This needs to be qualified by specifically stating the residue numbers in equivalent RNAP and citing a few structures they surveyed.

Since the location of σ 4 in the holoenzyme and in RPo is the same, it suggests that the interactions between σ 4 and RNAP are tight, possibly through the additional interactions with β CTH. These interactions restrict the conformational flexibility of σ 4, hence limiting the flexibility of spacer between -35 and -10, explaining the stringent requirement of the spacer. The additional interactions between M181 with specific bases could explain the sequence specificity. The tight interactions between σ 4 and RNAP, the specific read out of sequence of -35 and -10, and the unique unwinding mechanism would thus contribute significantly to the unique properties of ECF σ factors.

If possible, the authors should investigate what happens when β CTH is deleted and when both β CTH and β FTH are deleted in terms of spacer requirement. These should confirm the above points and make the manuscript more conclusive in providing molecular basis to the mechanisms and explanations to the unique properties of ECF σ .

Response to reviewers:

Reviewer #1 (Remarks to the Author):

The authors have adequately addressed my comments, except for one that requires some further clarification.

Regarding the response to comment 1 (experimental density maps): Is the “simulated Fo-Fc difference map” described in Figure S2A a simulated annealing omit difference density? If so, the authors should clarify this in the figure legend and explain exactly what was omitted – i.e. was only σ^H removed, but were nucleic acids still included? A similar clarification is needed for the “simulated Fo-Fc difference map” shown in Figures S6K, S7A/D, and 3E. If these are composite omit maps, please instead include that information.

After these changes are made, I recommend that the article be accepted for publication.

Reply:

Thanks for the suggestion. Yes, the Fig. S2A is a simulated annealing omit difference density with σ^H removed. The Figures S6K, S7A/D, and 3E show simulated annealing omit difference densities with nucleic acids removed. We have included such information in the respective figure legends.

Reviewer #2, comment 1 (Remarks to the Author):

The revised manuscript by Li et al. has addressed most of my concerns and I am happy for it to be accepted provided the following points are addressed/clarified.

Figure 2 and associated text on β CTH. The author needs to provide residue numbers for β CTH (both for *Mtb* and for *E. coli* numbering). The authors stated that “the interaction with β CTH was not observed in any of the previously reported bacterial sigmaA-RNAP structures”. This needs to be qualified by specifically stating the residue numbers in equivalent RNAP and citing a few structures they surveyed.

Reply: Thanks for the suggestion. The β CTH is not conserved in *E. coli* but is present broadly in many other bacteria. We have included into Supplemental Figure S2M a sequence alignment for RNAP β CTH of a few representative bacterial species. We also added residue numbers and citations for the surveyed structures in the main text.

Reviewer #2, comment 2 (Remarks to the Author):

Since the location of σ_4 in the holoenzyme and in RPo is the same, it suggests that the interactions between σ_4 and RNAP are tight, possibly through the additional interactions with β CTH. These interactions restrict the conformational flexibility of σ_4 , hence limiting the flexibility of spacer between -35 and -10, explaining the strident requirement of the spacer. The additional interactions between M181 with specific bases could explain the sequence specificity. The tight interactions between σ_4 and RNAP, the specific read out of sequence of -35 and -10, and the unique unwinding mechanism would thus contribute significantly to the unique properties of ECF σ factors.

If possible, the authors should investigate what happens when β FTH is deleted and when both β FTH and β CTH are deleted in terms of spacer requirement. These should confirm the above points and make the manuscript more conclusive in providing molecular basis to the mechanisms and explanations to the unique properties of ECF σ .

Reply:

Thanks for the suggestion. The β FTH is the anchor point for σ_4 of all bacterial σ factors and also the major anchor point for σ^H in our structures. Previous evidence has demonstrated that the β FTH is essential for the interaction of σ_4 to RNAP core enzyme and for the recognition the -35 element of promoter DNA. Removal of β FTH resulted in severely impaired transcription from -35-dependent promoters. Therefore, it is difficult to study the preference on -35/-10 spacers using β FTH-deleted RNAP.